# Neutron-encoded diubiquitins to profile linkage selectivity of deubiquitinating enzymes

Bianca D. M. van Tol [1], Bjorn R. van Doodewaerd[1],
Guinevere S. M. Lageveen-Kammeijer [2], Bas C. Jansen[2],
Cami M. P. Talavera Ormeño[1], Paul J. M. Hekking[1], Aysegul Sapmaz [1],
Robbert Q. Kim [1], Angeliki Moutsiopoulou[1], David Komander [3],
Manfred Wuhrer [2], Gerbrand J. van der Heden van Noort [1], Huib Ovaa [1,4] &
Paul P. Geurink [1] ✉

Deubiquitinating enzymes are key regulators in the ubiquitin system and an emerging class of drug targets. These proteases disassemble polyubiquitin chains and many deubiquitinases show selectivity for specific polyubiquitin linkages. However, most biochemical insights originate from studies of single diubiquitin linkages in isolation, whereas in cells all linkages coexist. To better mimic this diubiquitin substrate competition, we develop a multiplexed mass spectrometry-based deubiquitinase assay that can probe all ubiquitin linkage types simultaneously to quantify deubiquitinase activity in the presence of all potential diubiquitin substrates. For this, all eight native diubiquitins are generated and each linkage type is designed with a distinct molecular weight by incorporating neutron-encoded amino acids. Overall, 22 deubiquitinases are profiled, providing a three-dimensional overview of deubiquitinase linkage selectivity over time and enzyme concentration.

Ubiquitination is a post-translational modification (PTM) process in which ubiquitin (Ub), a highly stable 76 amino acid long protein[1], is covalently attached to a substrate protein to influence its function or location. Ub signalling is involved in almost all cellular pathways[2] and dysregulation has been observed in various diseases including various types of cancer and neurodegenerative diseases, metabolic disorders, and ageing[3]. Ub is installed via its C-terminal carboxylate, usually onto the amino group of the lysine side chain of a substrate protein, and this process is regulated by an enzymatic cascade involving a ubiquitin-activating (E1)[4], a ubiquitin-conjugating (E2)[5] and a ubiquitin-ligating (E3)[6] enzyme. Ub can also be attached to another Ub resulting in polyUb chains through the formation of an (iso-)peptide bond between the C-terminus of one Ub and the N-terminus (Met1) of the other Ub or one of its lysine sidechains (Lys6, Lys11, Lys27, Lys29,

Lys33, Lys48, and Lys63). As such, these polyUb chains come in eight homotypic linkage types (linked via the same (Lys) residue), but many more flavours exist[7,8]. Moreover, each polyUb linkage type has been found to result in different signalling functions[9]. And the installed Ub signals can be effectively antagonized by deubiquitinating enzymes (DUBs), a family of proteases that counteract the ubiquitination process by cleaving Ub from the target substrate protein or trimming polyUb chains[10-12].

Currently, approximately 100 different DUBs have been identified to be encoded in the human genome[11]. They are commonly divided into seven different families; ubiquitin carboxy (C)-terminal hydrolases (UCHs), ubiquitin-specific proteases (USPs), Machado-Joseph disease protein domain proteases (MJDs), ovarian tumour proteases (OTUs), JAB/MPN/Mov34 metalloenzyme (JAMM), motif interacting with

[1]Department of Cell and Chemical Biology, Chemical Biology and Drug Discovery, Leiden University Medical Center, 2333 ZC Leiden, The Netherlands. [2]Center for Proteomics and Metabolomics, Leiden University Medical Center, 2333 ZA Leiden, The Netherlands. [3]Ubiquitin Signalling Division, Walter and Eliza Hall Institute of Medical Research, 1G Royal Parade, Parkville, 3052 Melbourne, Victoria, Australia. [4]Deceased: Huib Ovaa. ✉e-mail: p.p.geurink@lumc.nl

Ub-containing novel DUB family (MINDY) and zinc finger with UFM-1 specific peptidase domain protein/C6orf113/ZUP1 (ZUFSP)[11]. DUBs are involved in many different cellular pathways, such as controlling proteasome-mediated protein degradation[2,11,13], DNA damage response[11,14–16], and innate immune signaling[11,17] and are implicated to be involved in different diseases[18–22]. For this reason, and because DUBs have potentially attractive druggable sites, these enzymes are recognized as promising drug targets[23]. Therefore, it is important to study DUB activity and elucidate their catalytic mode of action, efficiency, protein substrate preference, and ubiquitin linkage type selectivity.

As DUBs counteract the signal originating from a certain polyUb chain type, much effort has been dedicated to the determination of the linkage specificity of DUBs in the last decade[24–27]. Advances in synthetic strategies towards diUb[28,29] allowed the design of probes[30–32] and tools[33–35] targeting DUBs and provided insights into their molecular mechanisms. However, analysis mostly relied on incubation of a purified recombinant DUB with one diUb linkage-type at a time with an SDS-PAGE read-out (Fig. 1, top)[24,25]. Although advances have been made to allow a quantitative read-out by matrix-assisted laser desorption/ionization time-of-flight mass spectrometry (MALDI-TOF-MS)[26] or by fluorescence intensity[34] (Fig. 1, top), the element of natural competition between the different linkage types has so far been taken out of the equation.

While all polyUb linkages coexist in cells[7], it is currently unknown, and so far unaddressed, whether this coexistence influences general DUB activity, cleavage efficiency for each linkage as well as their linkage selectivity in general, as one or more linkage types can potentially influence DUB action on the other linkages. For example, questions arise whether there is direct competition between the linkage types, whether they influence the rate of each other's cleavage, and whether linkages are processed in a specific order when they are all present simultaneously. These are important questions to answer when trying to understand the different deubiquitination pathways in detail. However, the current assays are unable to answer these questions, since it is impossible to distinguish all linkages from each other during the read-out when they are present in a single mixture.

To overcome this, we developed an assay that allows the analysis of all different diUb linkages in a single mixture and with a single measurement. The diUb substrates were modified as little as possible, e.g. avoiding dyes, non-natural amino acids and other unnatural (MS) tags to obtain (near-) native diUb molecules. A complete set was designed that contained all diUb linkage types, seven isopeptide-linked diUb molecules (Lys6, Lys11, Lys27, Lys29, Lys33, Lys48, Lys63) and the linear diUb (Met1), with each isoform having a distinct mass effected by incorporation of fully $^{13}$C and $^{15}$N labelled amino acids. Accordingly, these diUb molecules can be measured all at once as MS can distinguish the identity and absolute amount of each linkage type present in the mixture (Fig. 1, bottom). Moreover, it allowed us to investigate Ub-linkage specificities of DUBs over time using a single mixture of all eight diUbs. Our screening method proved to be fast and repeatable and required only small amounts of diUb molecules. As a proof of concept, we assayed the selectivity and activity of 22 human

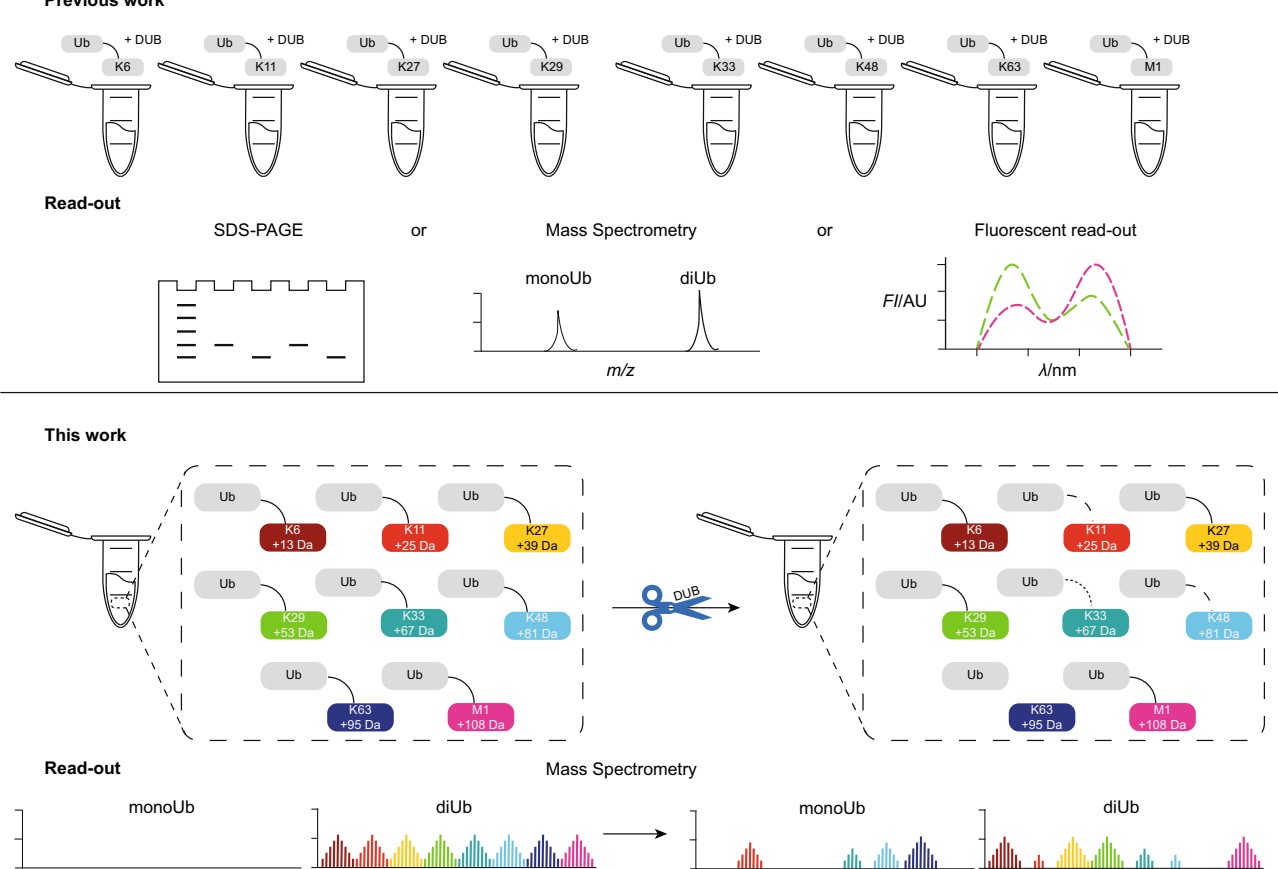

**Fig. 1 | The principle of the neutron-encoded diUb cleavage assay.** *Top*−Schematic outline of classical diUb cleavage assays relying on separate incubation of each diUb isoform with a DUB followed by SDS-PAGE, MS or fluorescent intensity read-out for every diUb isoform separately. *Bottom*−Schematic outline of the designed method where a DUB is incubated with a mixture containing all neutron- encoded diUb isoforms followed by MS analysis, allowing the quantification of each linkage from the complex mixture. Upon cleavage of the diUb by a DUB, the diUb signal(s) will disappear and the corresponding monoUb signal(s) will appear in the MS spectrum. Colors represent the differently linked diUbs as indicated.

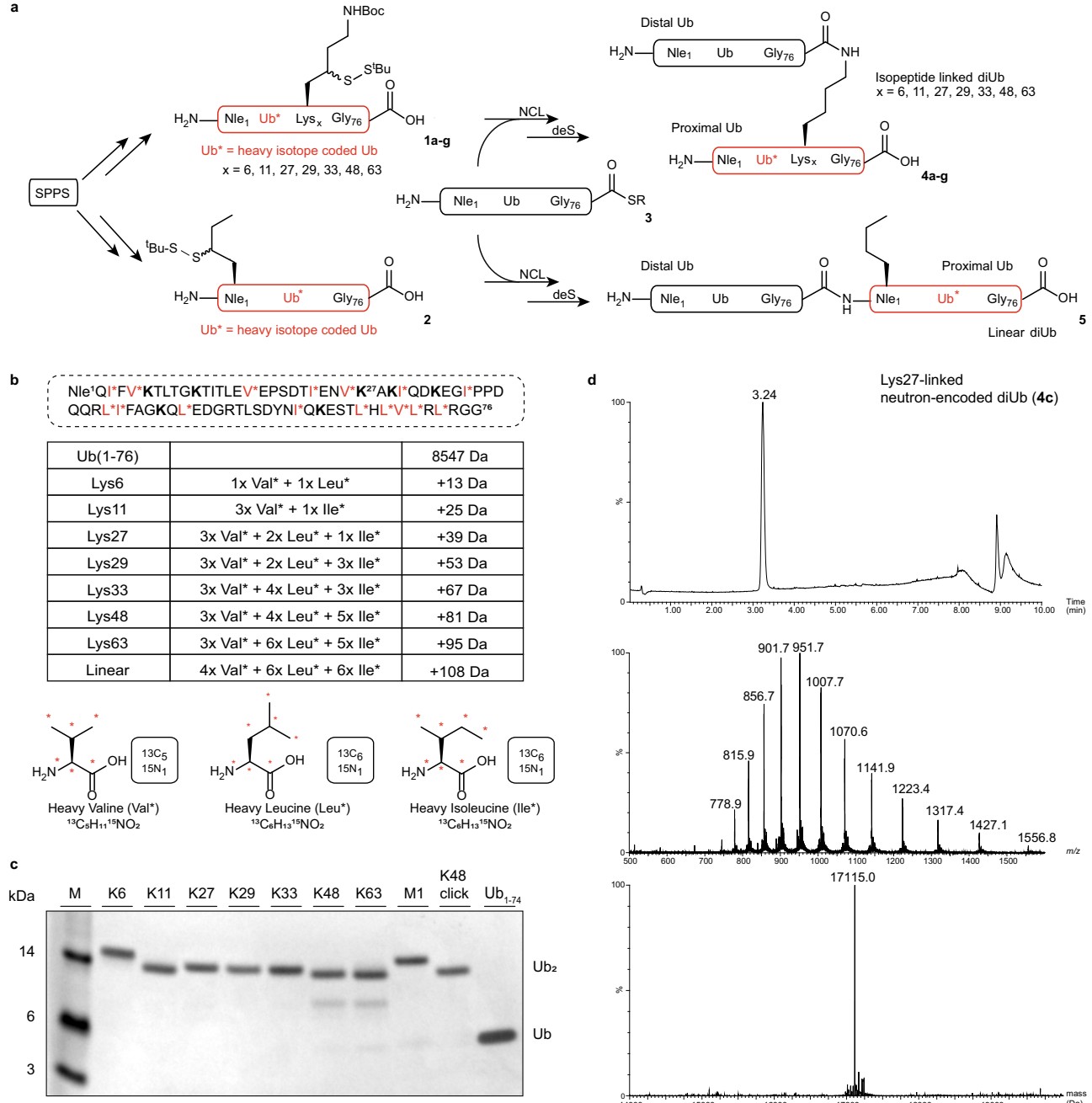

**Fig. 2 | Synthesis of all eight neutron-encoded diUbs. a** Synthesis scheme of neutron-encoded isopeptide-linked and linear diUbs using Solid Phase Peptide Synthesis (SPPS), Native Chemical Ligation (NCL) and radical desulfurization (deS). Reagents and conditions: Resin liberation and deprotection: 90.5% TFA, 5% H₂O, 2% TIS, 2.5% phenol, 6-40% yield; NCL: 0.1 M TCEP, 0.15 M MPAA, 6 M Gnd•HCl, 0.15 M sodium phosphate, pH 7.5, 37 °C; deS: 0.25 M TCEP, 0.1 M GSH, 0.075 M VA-044, 6 M Gnd•HCl, 0.15 M sodium phosphate, pH 7.0, 37 °C, 2.5-16% yield (over two steps). (Full synthetic scheme shown in Supplementary Fig. 1). **b** Labelling scheme of the eight diUb isoforms. Potentially heavy-isotope labelled amino acid positions in the

proximal Ub are *red* and marked with an asterisk. Nle Norleucine. Positions for linkage-dependent lysines to thiolysine replacements are shown in bold. The table shows the number of introduced neutron-encoded amino acids and the introduced mass difference for each linkage. The isotope labelled amino acids used are shown with the ¹³C and ¹⁵N atoms marked with *red* asterisks. **c** Coomassie-stained SDS-PAGE analysis of all eight neutron-encoded diUbs and used internal standards (*n* = 1). **d** Representative example of a total ion chromatogram (top), mass- (middle) and deconvoluted mass spectra (bottom) of Lys27-linked neutron encoded diUb (4c). Source data are provided as a source data file.

DUBs from different DUB families to obtain more insight into their proteolytic profile. Strikingly, we demonstrate that USP enzymes, while known for their chain type promiscuity, show linkage selectivity at lower enzyme concentrations; a phenomenon commonly observed for OTU family members[24]. In addition, we found that some USPs follow a consecutive cleavage order they start to process certain diUb linkages only after the consumption of other linkages has come close to completion.

## Results

### Design and synthesis of the neutron-encoded diUb molecules

The envisioned MS-based assay (Fig. 1, bottom) required the preparation of all eight diUb isoforms equipped with neutron-encoded amino acids to introduce appropriate mass differences (Fig. 2a, b). To ensure a suitable separation of isotopic patterns during MS analysis, a mass increase of 12, 13, or 14 Da for each consecutive diUb was chosen. Fully ¹³C and ¹⁵N labelled amino acids Val, Leu, and Ile were used to

introduce these mass differences because they are abundant in the ubiquitin sequence, introduce a substantial mass difference per amino acid (6 or 7 Da), and are relatively inexpensive. As the diUb linkage type is defined by the proximal Ub's lysine residue that forms the isopeptide bond, all neutron-encoded amino acids were incorporated in the proximal Ub only to directly link the appropriate mass fingerprint, as depicted in Fig. 2b and Supplementary Table 1, to the linkage type. This way each linkage could be identified by MS in both the diUb (substrate) as well as the monoUb (product) form (*vide infra*). Besides, since the assay relies on intact mass analysis only, the read-out is unaffected by the exact location of heavy-isotope introduction.

All eight diUb molecules were constructed using a native chemical ligation (NCL)-desulfurization (deS) strategy (Fig. 2a). First, all eight different proximal Ubs 1a-g and 2, harbouring the neutron-encoded amino acids and γ-thioLys[28,36–39] (in 1a-g) or γ-thioNle[40] (in 2) NCL handles, and distal Ub thioester 3 were synthesized via traditional Fmoc-based solid-phase peptide synthesis (SPPS) using an optimized version of our protocol (Fig. 2a and Supplementary Table 1)[28]. The NCL[28,34,38] of thiols 1a-g and 2 with thioester 3 followed by desulfurization under radical conditions[41] (Fig. 2a and Supplementary Fig. 1 and Supplementary Methods) and subsequent purification resulted in the eight neutron-encoded diUb molecules (4a-g and 5) in good overall yield (0.25–1.6 mg, 2.5–16%). The purity of the final products was analysed by SDS-PAGE (Fig. 2c and Supplementary Fig. 2) and LC-MS (Fig. 2d, Fig. 3a left and Supplementary Data. 1).

## All diUbs can be measured simultaneously and are processed by DUBs

To show that all eight neutron-encoded diUbs could be detected at the same time, they were mixed in an equimolar ratio and analysed by HPLC-MS. The obtained spectrum clearly showed a distinct isotopic pattern for each of the eight neutron-encoded diUb molecules and displayed a fair separation of all isotopic patterns at all charge states, thereby confirming that the introduced mass difference was sufficient to allow the detection of every single diUb linkage in the mixture (Fig. 3a right and Supplementary Fig. 3). To confirm that all neutron-encoded diUbs are correctly folded, and therefore accepted and processed by DUBs, all eight linkages were separately incubated for 3 hours with non-specific DUB USP21[25] and diUb proteolysis was analysed qualitatively by SDS-PAGE. This resulted in the observation of a cleavage pattern in line with reported data[25] (Fig. 3b and Supplementary Fig. 4). The diUb integrity of our synthetic constructs was confirmed in an assay where we compared the cleavage efficiency of neutron-encoded Lys48-, Lys63- and Met1-linked diUbs with their corresponding enzymatically prepared diUbs side-by-side using OTUB1 (reported Lys48 specific)[24], OTUD1 (reported Lys63 specific)[24] and USP21 (reported unspecific)[25]. DUB mediated hydrolysis was analysed by SDS-PAGE (Supplementary Figs. 5 and 6), which showed that the enzymatically prepared material and the synthetic material were processed comparably. Next, we investigated whether diUb cleavage and concomitant monoUb formation of different linkages could indeed be visualized by MS and whether (partly) processed and non-processed linkages could be distinguished from each other (Fig. 3c). To this end, OTUB2, a DUB known to selectively cleave Lys11, Lys48, and Lys63 linkages[24], was incubated with an equimolar mixture of the eight diUb molecules, and the reaction was followed over time by MS. As shown in Fig. 3c, we observed the disappearance of the expected diUb signals (Lys11, Lys48 and Lys63) over time and the appearance of the corresponding monoUb signals as well as the non-isotope coded distal monoUb (in grey). Interestingly, in contrast to reported findings, we also observed processing of other linkages after prolonged incubation (such as Lys6), which can be attributed to the unique features of our assay as we will address below in our quantitative analyses. The capability to measure all eight diUb

molecules simultaneously by MS and the OTUB2 cleavage results, illustrates that DUB selectivity can be measured in a mixture.

## Assay set-up and data analysis

The suitable diUb concentration range for the assay was determined by generating a standard curve of an equimolar mixture of all eight diUb molecules (0 – 2.0 μM). Absolute signals of three independent measurements were normalized to the internal standard non-hydrolysable triazole-linked Lys48 diUb[35] (non-hydrolysable clicked Lys48-linked diUb or K48 click, prepared using Copper(I)-catalyzed azide-alkyne cycloaddition (CuAAC)-chemistry) to calculate the measured concentration of diUb, which was plotted against the theoretical present diUb concentration. A linear response in concentration (and signal height) was well detected by the mass spectrometer between 0.5 μM and 2.0 μM of the diUbs and signals from diUb concentrations below 0.5 μM were less accurate compared to theoretically present diUb (Supplementary Fig. 7). Therefore, a starting concentration of 1.6 μM of each diUb linkage was chosen to ensure a reliable read-out throughout the entire assay time as the concentration of at least one of the analyte types, either monoUb or diUb, will always be above the linear detection threshold.

In the multiplexed assay, an equimolar mixture of all eight neutron-encoded diUbs (1.6 μM of each linkage) was incubated with recombinantly expressed and purified DUB (0.004 μM – 4.0 μM) at 37 °C. At different time points, small reaction samples were added to and therewith quenched by an acidic solution containing the internal standards and analysed by MS (Fig. 4a). $Ub_{1-74}$ and non-hydrolysable clicked Lys48 diUb[35] were chosen as internal standards to control for intrinsic HPLC-MS assay variances, such as injection volume and ionization variability, and to allow normalization and quantification. These controls were selected because their masses lie within the range of, but do not overlap with, the assay products (monoUbs) and substrates (diUbs) respectively.

A big advantage of our assay is the possibility to perform the MS measurements using an intact mass measurement approach, which makes the preparation of the MS samples easy with minimal loss of material. Since each diUb isoform has a distinct ionization pattern, thereby impeding single-charge-state quantification[42–45], we performed quantification over the whole charge state range of the proteins ($z=10^+$ to $z=25^+$ for diUb and $z=5^+$ to $z=13^+$ for monoUb) using a tailor-made version of the open-source software package LaCyTools[46,47].

To confirm the validity of quantification with LaCyTools, different datasets were explored using the LaCyTools software and compared to a manual analysis (see Supplementary Fig. 8 for details). This revealed that differences between automated and manual analyses were small and the ratio between all analytes within each dataset was comparable.

From the recorded data, diUb disappearance, as well as monoUb appearance, can be quantified, and these values should correspond with each other (Fig. 4c). The percentage of consumed diUb substrate and the concentration of formed monoUb were calculated and plotted over time for all measured DUBs at different concentrations (Supplementary Data 2a–c). DUB assay results are summarised in heat maps (Fig. 5a, b, and Fig. 6a) where the amount of consumed diUb substrates after 180 min is shown for different DUBs and at multiple enzyme concentrations. For data interpretation, both diUb consumption and monoUb appearance were taken into account.

## Determining DUB specificity during linkage competition

We applied our MS DUB assay (Fig. 4a) to determine the linkage specificities of 22 recombinant human DUBs, divided over three groups (Fig. 4b) under linkage competition conditions. The enzymatic reactions were performed using four different DUB concentrations. We started with seven well-known and well-studied DUBs from the panel used for Ubiquitin Chain Restriction analysis (UbiCRest), OTUB1,

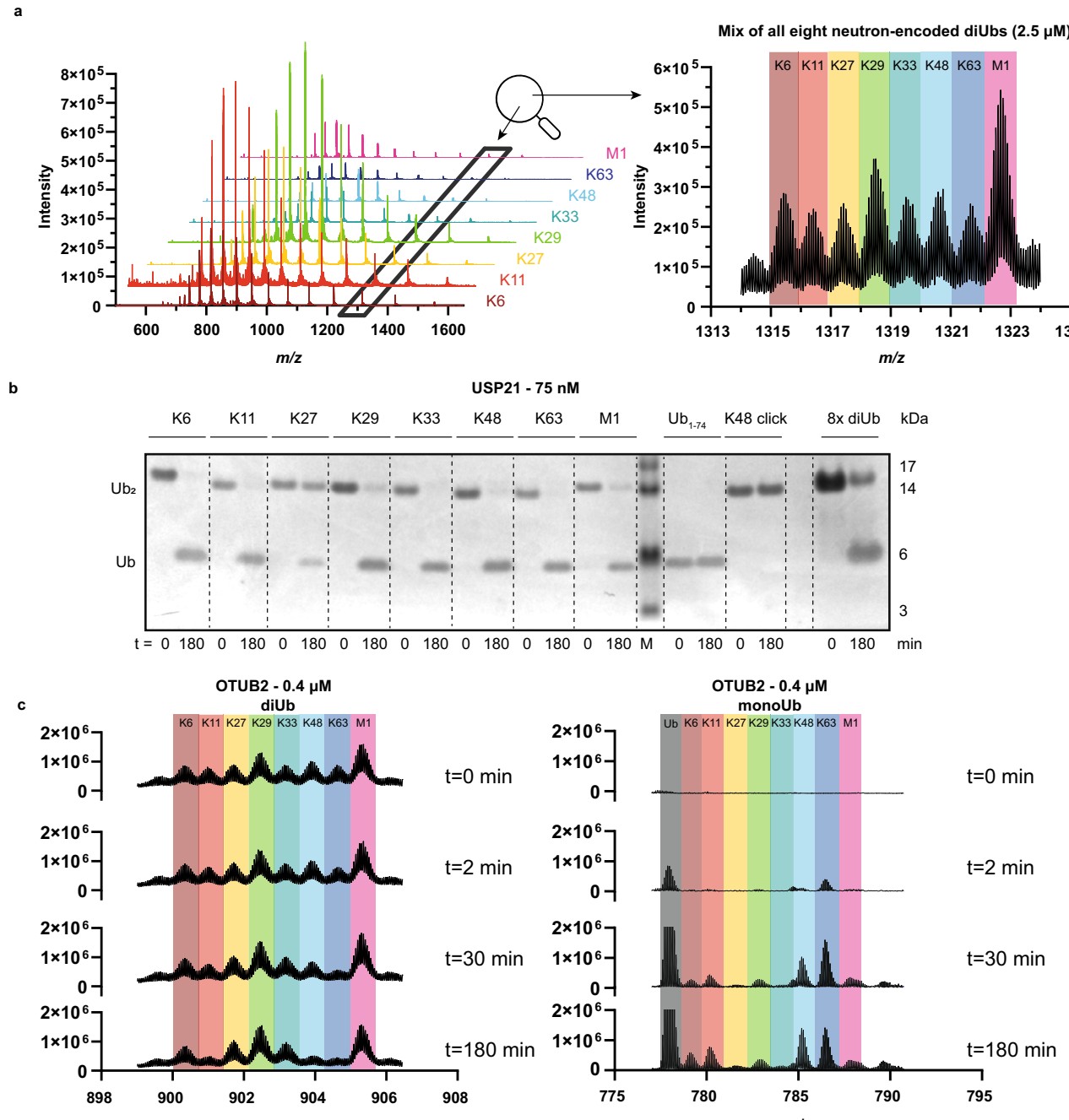

**Fig. 3 | Proof of Principle Assay. a** Overlayed mass spectra of all eight neutron-encoded diUbs (4a-g and 5) measured separately (left panel) and measured in a single analytical LC-MS run in equimolar amounts, magnified at charge state $z = 13^+$ (right panel). Colors represent the differently linked diUbs as indicated. **b** SDS-PAGE analysis of USP21 mediated proteolysis of all eight neutron-encoded diUb. All eight neutron-encoded diUbs (2.5 μM each), internal standards and a mixture of all eight neutron-encoded diUbs (8x 1.6 μM) were incubated with USP21 (75 nM) at 37 °C ($n = 1$). **c** OTUB2 (0.4 μM) mediated cleavage of all eight diUbs (1.6 μM each), analysed by MS at indicated time points. Charge states $z = 19^+$ and $z = 11^+$ are shown for diUb and monoUb, respectively. MonoUb signal in grey represents the sum of all released distal monoUbs. Source data are provided as a source data file.

OTUB2, OTUD2, OTUD3, OTULIN, Cezanne and USP21, to check whether the data produced in our assay corresponds to literature[48]. These seven DUBs are reported to be specific towards one linkage or a subset of linkages. Using our assay we were able to confirm their known specificities (Fig. 5a and Supplementary Data 2a). OTUB1 showed a preference for Lys48-linked diUb after 180 minutes of processing at different concentrations (Fig. 5a), which is its reported preferred linkage[24,48]. However, the full cleavage profile (Supplementary Data 2a) also showed processing of Lys63-linked chains at 0.4 and 4.0 μM and monoUb products from Met1-linked chains were observed

at high concentration (4.0 μM). OTUB2 at low concentration (0.04 μM) cleaved the preferred Lys63 linkage but broadened its preference to include Lys11 and Lys48 at moderate enzyme concentration (0.4 μM)[24]. Notably, at 4.0 μM the specificity further broadened to include cleavage of Lys6 diUb, a so-far unreported observation[24,26,27]. OTUD2 only showed activity at 4.0 μM with a main preference for Lys11 diUb (Fig. 5a). OTUD3 specifically cleaved Lys6 and Lys11 diUb at 0.4 μM, confirming literature observations[24,48,49] but interestingly also showed nearly full conversion of Lys48 and Lys63 diUb at 4.0 μM. For OTULIN we observed an exclusive consumption of the Met1-linked diUb even at

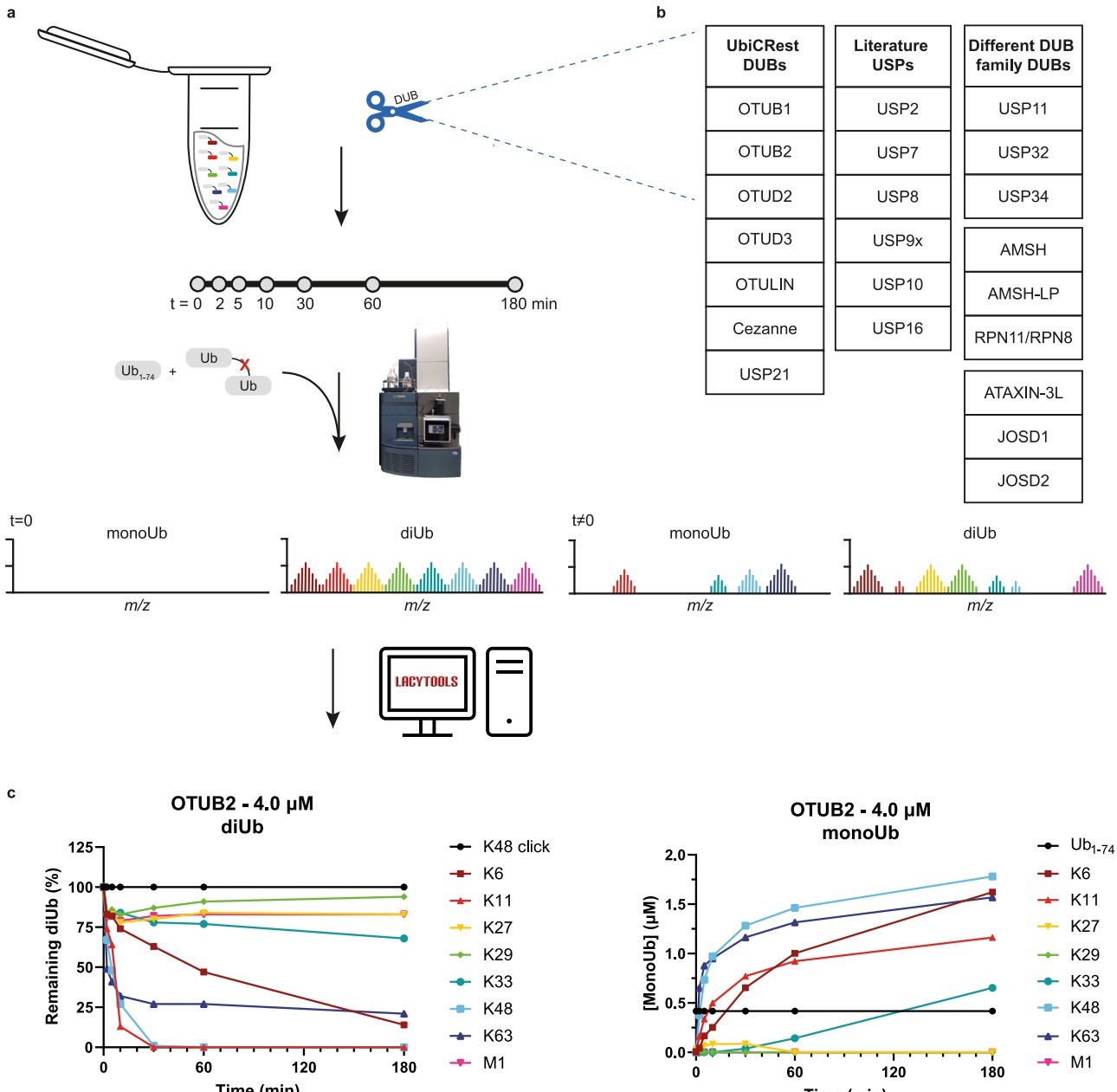

**Fig. 4 | Workflow of the neutron-encoded diUb cleavage assay. a** Schematic representation of the DUB assay and MS analysis. A DUB is incubated with a mixture containing all eight diUb molecules. At the indicated time points, a sample of the reaction mixture was taken and quenched by diluting in an acidic solution containing the internal standards. Subsequently, the samples were analysed by LC-MS and the data was processed using LaCyTools software. **b** The 22 selected DUBs that were analyzed with the assay. **c** Quantification of OTUB2 mediated diUb cleavage in a mixture containing all eight diUbs (1.6 μM each), represented by diUb disappearance (left) and monoUb appearance (right) (*n* = 1). Colors represent the differently linked diUbs as indicated. Source data are provided as a source data file.

the highest enzyme concentration, corroborating that this is an M1 specific protease[48,50]. Cezanne specifically cleaved Lys11-linked diUbs at lower concentrations, but loses this specificity at higher enzyme concentrations[24], as it processed Lys6-, Lys48- and Lys63-linked diUbs at 0.4 μM and 4.0 μM as well. USP21 is reported to cleave all diUb linkages[48,51], which is indeed confirmed by our data. Of note, generally Lys27-linked diUb is often not (fully) processed. Our data confirms that this is not due to incorrect folding of Lys27-linked diUb, since proper processing of this linkage was observed at a high concentration of USP21 (4.0 μM).

Furthermore, we investigated six well-studied DUBs from the USP family, which are known for their chain-type promiscuity (USP2, USP7, USP8, USP9x, USP16, and USP10) (Fig. 5b and Supplementary Data 2b).

We found that USP2 mainly cleaved Lys11 diUb at low enzyme concentration (0.04 μM) (Fig. 5b) and consumption of this linkage was also faster than other linkages at 0.4 μM of USP2 (Fig. 5c). USP7 showed a preference for Lys6, Lys11, Lys33, Lys48, and Lys63 diUb although processing of Lys33 diUb was slower compared to the other linkages (Supplementary Data 2b). At 0.4 μM USP8 showed a similar specificity pattern as USP7 at 0.04 μM. USP9x cleaved Lys6, Lys11, Lys48, and Lys63 diUb at 0.4 μM, but at a higher concentration cleavage of Lys29, Lys33, and Met1 diUb was also observed (Fig. 5b). Strikingly, USP9x only started the consumption of Lys29 and Lys33 chains after Lys6, Lys11, Lys48 and Lys63 were almost fully processed (Fig. 5c in combination with Supplementary Data 2b). USP16 showed a similar pattern as USP21, with apparent linkage-type promiscuity. USP10 showed a preference for

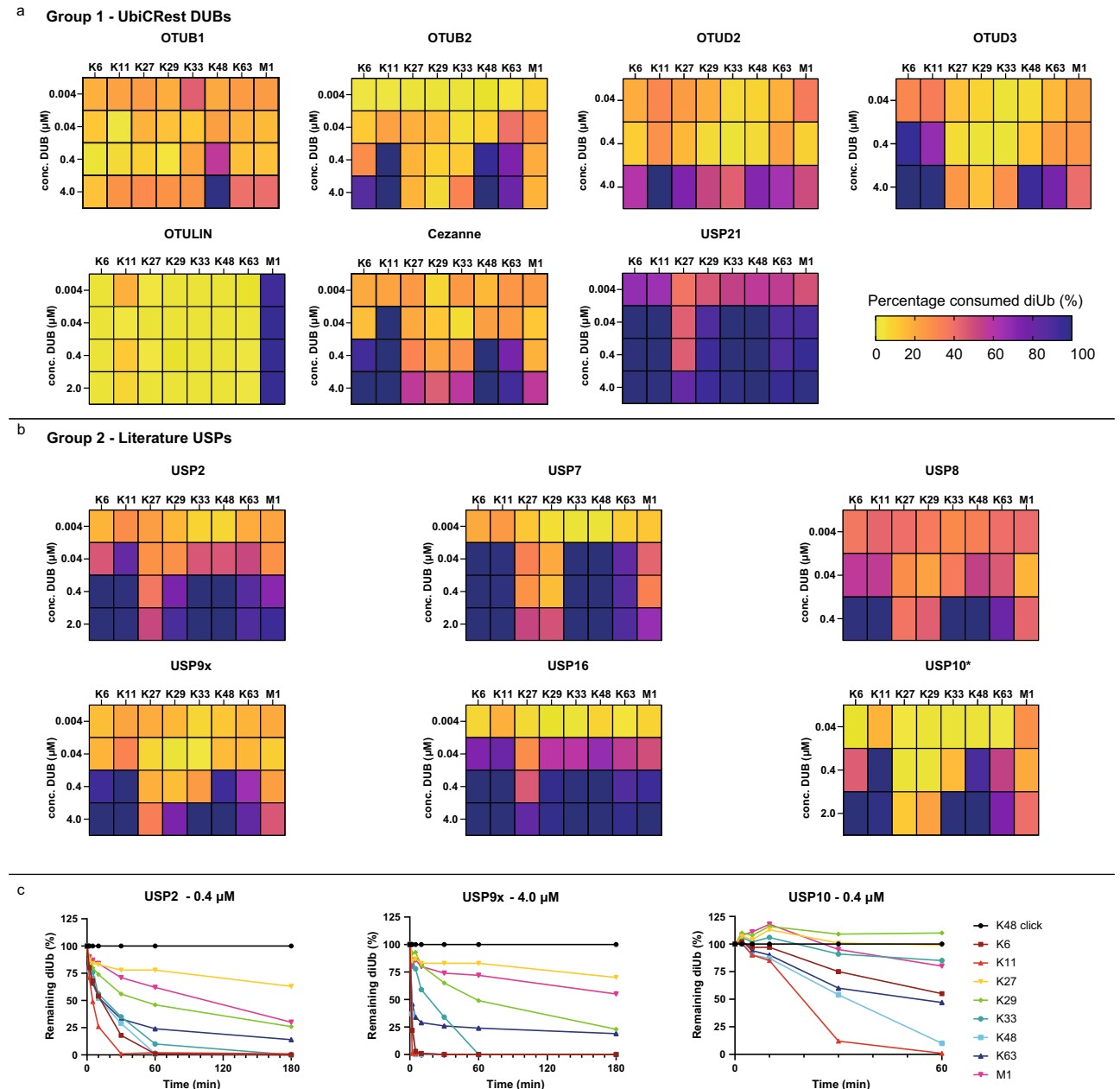

**Fig. 5 | Linkage specificity profiles of well-studied OTU and USP DUBs.** Heatmaps showing the percentage of consumed diUb isoforms per DUB of **a** UbiCRest panel or **b** USP family at different concentrations after 180 min. Asterisk (*) indicates that the heatmap of USP10 was constructed with diUb concentration after 60 min. **c** Full quantification profiles of diUb disappearance over time for USP2 (0.4 μM), USP9x (4.0 μM) and USP10 (0.4 μM). An equimolar mixture (8 x 1.6 μM), containing all neutron-encoded diUb isoforms, was incubated with the DUB at 37 °C, samples were taken at the indicated time points and analysed by LC-MS (*n* = 1). Colors represent the differently linked diUbs as indicated. For full DUB characterization profiles see Supplementary Data 2a, b. Source data are provided as a source data file.

Lys11 and Lys48 diUb and to a lesser extent for Lys6 and Lys63 diUb at lower DUB concentrations (0.4 μM). At 2.0 μM Lys33 diUb was also fully converted but processing of this chain appeared to start only once Lys11 and Lys48 diUb processing had come close to completion (Fig. 5c in combination with Supplementary Data 2).

Next, we investigated the specificity of nine DUBs from different DUB families (Fig. 6a) at a single DUB concentration of 4.0 μM. Some of these DUBs were taken along in reported specificity screens and usually, results were analysed qualitatively at only one timepoint using SDS-PAGE analysis. So, we set out to complement their selectivity profiles in a quantitative manner using our assay. USP11 processed Lys6, Lys11, Lys33, Lys48, and Lys63 diUb (almost) completely after 180 min, while Lys29 diUb was only partially processed (Fig. 6a, b, and

Supplementary Data 2c). Similar results were obtained for USP32, although Lys29 was processed to a higher extent after prolonged incubation times, and also Lys27 and Met1 processing was observed. USP34 showed low activity with a slight preference for Lys63 chains.

The metallo-DUBs AMSH and AMSH-LP were moderately active at 4.0 μM and specifically processed Lys63 diUb as expected[26,52–54]. The yeast RPN11/RPN8 complex, which is comparable to the human PSMD14/PSMD7 complex, has previously been reported to hydrolyse all isopeptide-linked diubiquitins[55]. In our assay we found a main preference for Lys11 diUb with almost full proteolysis after 3 h, although other linkages are also processed to lesser extent.

The three enzymes from the MJD family (ATAXIN-3L, JOSD1, and JOSD2) displayed low activity, even at 4.0 μM, which made it difficult to

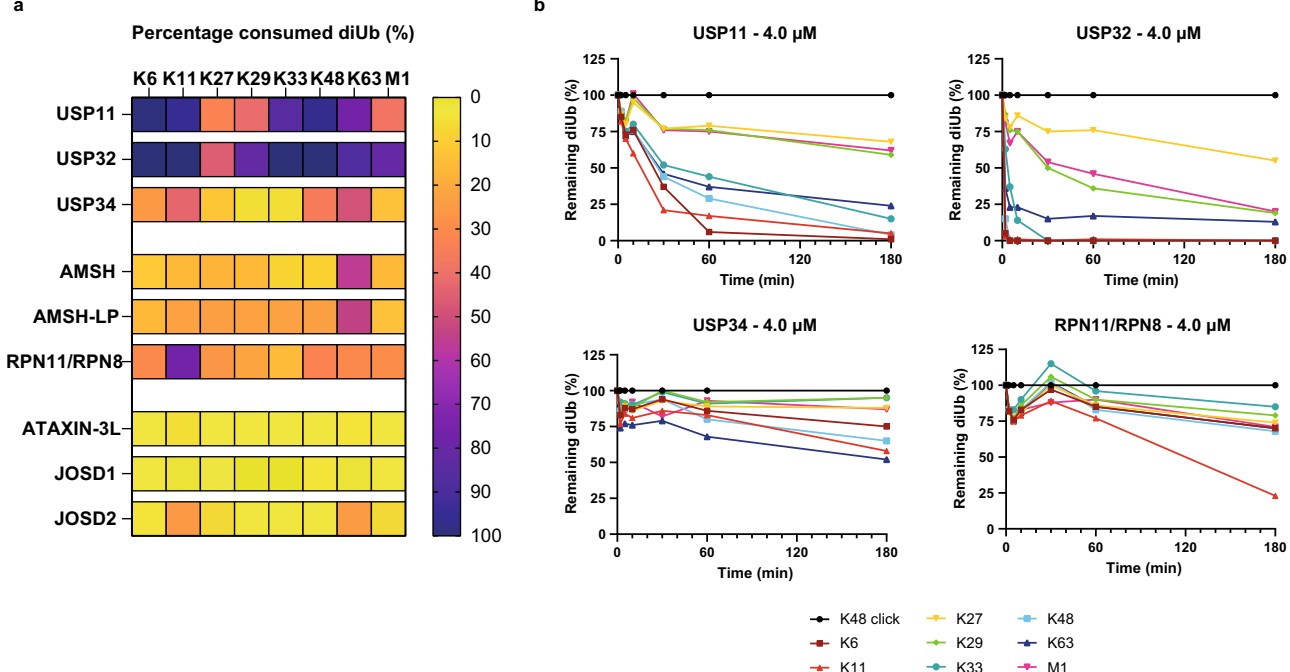

**Fig. 6 | Linkage specificity profiles of DUBs from different DUB families.**
**a** Heatmap showing the percentage of consumed diUb isoforms by 9 different DUBs (4.0 μM) from different DUB families after 180 min incubation. **b** Full quantification profiles of diUb disappearance over time for USP11, USP32, USP34 and RPN11/RPN8. An equimolar mixture (8 x 1.6 μM), containing all neutron-encoded diUb isoforms, was incubated with the DUB at 37 °C, samples were taken at the indicated time points and analysed by LC-MS (n = 1). Colors represent the differently linked diUbs as indicated. For full DUB characterization profiles see Supplementary Data 2c. Source data are provided as a source data file.

properly profile these DUBs with our assay. No clear substrate conversion could be observed in the diUb channel, except for JOSD2, the most active DUB from the family, that processed some Lys11 and Lys63 chains (Fig. 6a, b, and Supplementary Data 2c), which is in line with earlier findings[56,57].

## Discussion

Given the critical roles of DUBs in a lot of cellular processes, there is a growing interest to exploit them as targets for drug development[58]. Further establishing insights into their functioning at a specific location in the cell or their activity under certain circumstances are important additional steps to couple DUB activity with therapeutic intervention. Profiling their linkage selectivity and likewise, their preference when a variety of substrates, for example, different polyUb linkages, are at their disposal is important in such efforts. Our fast, adaptable and quantifiable assay to monitor a DUB's proteolytic preference, activity and selectivity in vitro, when all eight natural diUb linkages compete with each other, adds to the growing toolbox of assays and dedicated reagents to decipher part of the Ub network. The key conceptual advantage of our MS assay lies in its ability to analyse DUB specificity in the presence of all diUb linkage types in a single assay. This was achieved by synthesizing a set of all eight native diUbs, in which each linkage type has its own fully $^{13}$C and $^{15}$N labelled amino acid fingerprint, resulting in eight diUb isoforms with a distinguishable molecular weight. Altering the substrate or enzyme concentration, or potentially the influence of an allosteric binder (e.g. another Ub linkage) can be easily measured and quantified on a mass spectrometer in a measure of minutes, compared to longer periods when several SDS-PAGE assays need to be performed side by side[24,25].

Isotope coding is a commonly used technique in MS-based analysis methods, such as tandem mass tag (TMT) labelling[59], stable isotope labelling by amino acids in cell culture (SILAC)[60] or the absolute quantification strategy (AQUA)[61], to identify or comparatively quantify specific proteins in a complex biological sample. However, none of these methods can (easily) be applied in a straightforward and fast DUB assay, without comprising the structural integrity of the Ub chains. Since by using conventional techniques the differential labelling of all eight diUb linkages can only be achieved by the instalment of artificial MS tags in the substrates. These small but intrinsic differences between the resulting diUb linkage types might impact DUB recognition and/or processing parameters. Furthermore, most isotope coding techniques require substantial sample preparation, e.g. tryptic digestion or peptide enrichment, which is time-consuming and can result in sample loss. Many of these methods require tandem MS analysis, which can be problematic with analytes that have an identical retention time. Our method on the other hand, in which we take full advantage of chemical protein synthesis in combination with isotopically labelled amino acids, bypasses these issues. The sophisticated stable-isotope code we installed, allows for proper baseline separation in the mass spectra and hence discrimination between each of the eight diUb analytes and nine monoUb analytes, without complex sample processing or modifying any of the biophysical properties of the diUbs apart from their molecular weight. The only alteration we made was the replacement of Met1 by its bioisostere norleucine (Nle) to avoid the notorious oxidation of the Met1 thioether moiety, leading to different molecular weights (M, M +16Da and M+32Da) for a single type of Ub protein, which is detrimental to MS analysis and quantification. This generally accepted Met1Nle substitution typically does not affect recognition by DUBs[40,62].

In this study, we demonstrated that, by using an intermediate resolution mass spectrometer, we can already achieve sufficient resolution and sensitivity which makes it an attractive workflow that could be incorporated in other (bio)chemical laboratories. Moreover, we expect that the sensitivity of our assay will even be further improved with the continuous development of new technologies in the field of mass spectrometry, that will drive the limit of detection and improvement of resolution to an even higher level in the future. Eventually, this would lead to lower amounts of required substrates to

be detected, thereby making it easier to switch from equimolar ratios between diUb linkages to ratios that are found in cells, which will bring the assay one step closer to in vivo circumstances. The possibility to detect lower concentrations could even allow for Michaelis-Menten kinetic measurements in a linkage competitive setting.

Data analysis is an important part of our assay and although it is possible to perform manually, it is a laborious process that is prone to errors. By adjusting LaCyTools, an open-source software package that was initially developed in-house to analyse glycopeptides[46,47], we managed to streamline our data analysis. LaCyTools automatically aligns, calibrates, and integrates LC-MS data with the appropriate quality control (e.g. ppm error, S/N and isotopic pattern comparison), and in addition allows to fully define the atomic composition of analytes, including our neutron-encoded (di)Ubs. To further simplify data analysis we implemented two essential features in our assay. First, the proteins were eluted using a shallow HPLC gradient to separate monoUb and diUb at baseline level, which prevents an undesired overlap of the monoUb and diUb MS signals. Second, we make use of two internal standards, $Ub_{1-74}$ and non-hydrolysable Lys48 diUb, which have a molecular weight in the same mass range as the neutron-encoded monoUbs and diUbs, respectively, to account for ionisation efficiency differences due to protein size. This allowed us to properly quantify both the diUb consumption as well as the monoUb formation, which led to some important observations detailed below.

In some cases, the observed diUb consumption does not match the monoUb formation. Illustrative of this is the data for USP2 (0.4 μM) (Supplementary Data 2b), which has a clear preference for K11 diUb, followed by K6, K33, K48 and K63 diUb in the diUb channel, whereas the monoUb channel shows the highest formation of K29-derived monoUb. A concomitant observation is that upon full conversion in the diUb channel, the end-point signal in the monoUb differs for each linkage, sometimes reaching values above the initial substrate concentration of 1.6 μM, which is for example clearly shown for USP16 at 4.0 μM (Supplementary Data 2b). A possible explanation lies with the equilibration of the diUb mixture, which was based on quantification of the protein bands after SDS-PAGE analysis and Coomassie staining (like Fig. 2c) as an orthogonal method with respect to MS. Although this is a generally accepted method, the actual diUb concentrations can slightly differ, which will eventually affect the calculated concentration of formed monoUb. Another important point is the data processing by LaCyTools, where we applied very stringent quality control and background subtraction to our MS data. This can affect the quantification and is most pronounced for lower MS signals, which is well in line with our determination of the optimal substrate concentration to be above 0.8 μM. Overall, the most accurate linkage specificity data is obtained by considering both the diUb and monoUb channels, which can easily be done from a single measurement in our assay.

Overall, the results obtained with our neutron-encoded diUb substrates were in good agreement with literature[24–27,48–53,55,56] but also provided some interesting insights into DUB selectivity. OTUB1 has previously been annotated as Lys48 specific[24], but we also observed cleavage of Lys63- and Met1-linked chains (Supplementary Data 2a). Although the processing of Lys63-linked linkages has previously been observed[63], the processing of Met1-linked chains was only recently described for OsOTUB1, a homologue of human OTUB1[64]. We found OTUB2 to also process Lys6 diUb but only at elevated enzyme concentration and upon almost full consumption of its known target linkages Lys63, Lys48, and Lys11 (Fig. 5a and Supplementary Data 2a). The OTU domain of OTUD3 beautifully illustrates a dose-dependent specificity change (Fig. 5a and Supplementary Data 2a), where at low enzyme concentration it preferentially cleaved Lys6 followed by Lys11 chains but at elevated enzyme concentration, its specificity is broadened to include Lys48 and Lys63 diUb after (almost full) consumption of Lys6 and Lys11 linkages. The observation that the

linkage specificity changes with DUB concentration has been well documented for the OTU DUB family[24], but this has so far not been observed for USP DUBs. We here demonstrated a concentration-dependent linkage specificity for some members of the USP family. For example, USP2 which initially processes Lys11 diUb at 0.04 μM, additionally processes Lys6, Lys29, Lys33, Lys48, Lys63, and Met1 diUb at 0.4 μM and cleaves all diUbs except the Lys27 linkage equally fast at 4.0 μM (Fig. 5b and Supplementary Data 2b). USP7 also showed a pronounced linkage selectivity pattern at intermediate enzyme concentrations which became less prominent at high enzyme concentrations (Fig. 5b and Supplementary Data 2b). Furthermore, USP9x and USP10 cleavage patterns pointed towards a preference in cleavage order for the different processed diUbs; they only start processing certain linkage types after other linkage types are almost fully consumed by the enzyme first (Supplementary Data 2b). These results showcase the potential of our assay, where these USPs were reported to lack a clear linkage specificity, our data showed that under linkage competition conditions their assumed chain type promiscuity is aberrant. Notably, since the reaction progress is being monitored at several timepoints (Fig. 4a), it is possible to deduce information on the cleavage rates of each diUb linkage type relative to each other. This provided some interesting observations that differ from literature findings. USP9x was earlier shown to faster process Lys63- compared to Lys11-linked diUb[65], whereas our data showed the exact opposite effect (Supplementary Data 2b). Kinetic analysis of USP7 previously revealed a 1.5 times higher catalytic efficiency towards Lys33- over Lys11- and Lys63-linked diUb[25], which contrasts our finding that Lys33 is processed more slowly compared to Lys63 and Lys11 (Supplementary Data 2b). Also, our data revealed an similar processing rate of Lys48- and Lys63-linked diUb by Cezanne (Supplementary Data 2a), where a fourfold higher catalytic efficiency towards Lys63- over Lys48-linked diUb was reported previously[66]. These observed differences in hydrolysis rates between those reported for individual chains and the ones we found in the mixture containing all linkages may indicate that the presence of certain diUb linkages can lead to a change in hydrolysis rates of other linkages.

Another interesting finding is the high preference of the metallo-DUB RPN11/RPN8 complex for Lys11 diUb. The RPN11/RPN8 complex is reported to process all linkages, however the catalytic efficiency for the processing of Lys11-linked diUb is two times higher as for Lys48-linked diUb and four times higher as for Lys63-linked diUb[55]. In accordance with the reported catalytic efficiencies, we observed efficient processing of Lys11-linked diUb after 180 minutes, while only half of the amount of Lys48-related monoUb was formed and even less Lys63-related monoUb (Fig. 6 and Supplementary Data 2c).

We identified several cleavage patterns of DUBs that were not consistent with those obtained from conventional cleavage assays. Since our assay possesses the element of linkage competition, which is also present in cells, the selectivity pattern of DUBs can potentially be influenced by the presence of one or more diUb linkages. Our data will therefore provide further insights into the physiological role of these DUBs.

We believe that our method provides a major progress in the field of characterizing Ub linkage specificities of DUBs but at the same time the method in its current state is only limited to diUb linkages. Although the use of diUb tools has been widely considered a valid approach, it does not address the full complexity of Ub biology. In cells, more complex polyUb chains, including higher order homotypic and heterotypic chains, branched chains and hybrid chains of Ub with Ub-like proteins exist, and these chains are often conjugated to substrate proteins. DUB activity can be affected by the existence of such chains as is for example well illustrated by the finding that the DUB MINDY-1 only processes tetraUb or longer Lys48-linked chains[67]. The ability to include more complex Ub chains beyond diUb would therefore be a valuable extension of our method. The nature of our

assay, the widely available chemical preparations of Ub conjugates, as well as our MS-based read-out, should in principle allow for its application in the analysis of more complex Ub chains. The main determinant is the possibility to chemically introduce sufficient mass diffences to discriminate between each individual Ub species. Current synthetic procedures already allow for the preparation of (branched) triUb, tetra-, penta-, hexa-Ub and ubiquitinated peptides in a controlled way starting from SPPS[36,68-70]. Our MS-based assay set-up can easily be multiplexed further beyond the currently used 8-plex format. For example, with Ub trimers a total mass difference of 600 Da can be introduced before the charge states begin to overlap. With the 12 Da mass difference for every consecutive Ub, an up to 50-plex MS assay would theoretically be possible. The appropriate mass differences can be achieved by the introduction of more and different neutron-encoded amino acids via SPPS and this is possible on nearly any location in the Ub sequence since all fully $^{13}C,^{15}N$-labelled amino acids are commercially available. Ongoing developments in Ub synthesis may enable the preparation of even more complex Ub chains in the future, but with the currently available methods we anticipate that our assay, can be extended to all tri- and tetraUb isoforms, which will already mean a substantial step forward towards complex Ub chains present in cells.

In summary, we present an MS-based assay to profile the Ub linkage specificities of deubiquitinating enzymes (DUBs) when all eight existing ubiquitin linkage types are present in a single mixture. This was achieved by synthesizing a neutron-encoded set of eight native diUb isoforms, each having a distinguishable molecular weight. Using our assay we profiled the linkage specificity of 22 DUBs, which led to insights into their mode of action that could not have been acquired with conventional methods. Our data revealed that direct competition between different diUb linkages can alter a DUBs Ub chain specificity pattern and we provide tentative evidence that different diUb linkages are sometimes processed in a specific order. The assay has several advantages over existing methods: it makes use of the isotopologues of native diUb substrates, is quantitative, requires low amounts of material, the read-out can be done in a mid-throughput fashion, and competition between linkages is taken into account. The straightforward nature of the experimental setup makes the assay easily adaptable and custom-tailored for a biochemical lab having access to a mass spectrometer. We hence expect our neutron-encoded diUb assay to become invaluable in analysing DUB linkage specificity and anticipate its application in different studies under different circumstances to help shine a light on the intricacies of deubiquitination.

## Methods

### Fmoc protection of neutron-encoded amino acid
Amino Acid (AA; $^{13}C_x$, $^{15}N_1$) (1 eq.) was dissolved in 10% $Na_2CO_3$ in $H_2O$ (11 mL/mmol AA) and the solution was cooled to 0 °C. FmocOSu (1.2 eq.) was dissolved in 1,4-dioxane (7 mL/mmol FmocOSu). The FmocOSu solution was added dropwise to the cooled amino acid solution over the course of 2h. The reaction mixture was allowed to warm to room temperature (rt) and was stirred for 16 hrs. $H_2O$ (11 mL/mmol AA) was added to the reaction mixture resulting in a clear solution. The reaction solution was washed with $Et_2O$ (3x 30 mL/mmol AA). The aqueous layer was acidified to pH ~1 with conc. HCl and extracted with EtOAc (2x 16mL/mmol AA and 1x 30 mL/mmol AA). The combined organic phase was washed with BRINE (2x 25 mL/ mmol AA), dried over $Na_2SO_4$ and concentrated *in vacuo*. The resulting residue was purified by Büchi flash column chromatography (100% n-Hept → 100% EtOAc) to yield the pure Fmoc-protected neutron-encoded amino acid. (Supplementary Fig. 9).

### Solid-phase peptide synthesis of Ub polypeptides
All Ub polypeptides were synthesized by standard 9-fluorenylmethoxycarbonyl (Fmoc)-based linear SPPS on 2-chloro trityl resin or HMPA resin[28]. Detailed procedures for the synthesis can be found in the Supplementary Information – Supplementary Methods.

### Synthesis of diUb substrates
Monoubiquitins 6a-g, 7 and 8 on resin were synthesized using solid-phase peptide synthesis (SPPS). Monoubiquitins 7a-g were liberated from the resin and deprotected using TFA/$H_2O$/PhOH/$i$Pr$_3$SiH (90.5/5/2.5/2; v/v/v/v), yielding neutron-encoded Ub$_{1-76}$ containing γ-thioLys 1a-g. Monoubiquitin 7 was liberated from the resin using mild acidic conditions (20% HFIP/$CH_2Cl_2$; v/v) while protecting groups on the amino acid side chains remained intact. Methyl 3-(glycylthio)propionate was coupled to the liberated C-terminal glycine yielding 9 using EDC and HOBt in DCM. Acid-mediated deprotection yielded Ub$_{1-76}$-thioester 3 in multi milligram amounts. Resin-bound monoubiquitin 8 was elongated with γ-thioNle using the SPPS coupling conditions resulting in resin-bound monoubiquitin 10. Neutron-encoded Ub$_{1-76}$ containing γ-thioNle 2 was obtained after resin cleavage and amino acid side chain global deprotection using TFA/$H_2O$/PhOH/$i$Pr$_3$SiH (90.5/5/2.5/2; v/v/v/v). Native chemical ligation (NCL) reactions between Ub$_{1-76}$-thioester 3 and neutron-encoded Ub$_{1-76}$ containing γ-thioLys 1a-g or neutron-encoded Ub$_{1-76}$ containing γ-thioNle 2 yielded neutron-encoded diUb 11a-g and 12. Finally, the remaining sulfur atom was removed using desulfurization under radical conditions[41] to obtain the native diUb molecules 4a-g and 5 (Supplementary Fig. 1).

### Synthesis of internal standard non-hydrolyzable clicked Lys48 diUb
Monoubiquitins 19 and 20 on resin were synthesized using linear solid-phase peptide synthesis (SPPS). Monoubiquitin 19 was liberated from the resin using mild acidic conditions (20% HFIP/$CH_2Cl_2$; v/v) while protecting groups on the amino acid side chains remained intact. Propargylamine was coupled to the liberated C-terminal glycine followed by acid-mediated deprotection yielded Ub$_{1-75}$-PA 21. Monoubiquitin 20 was liberated from the resin and deprotected using TFA/$H_2O$/PhOH/$i$Pr$_3$SiH (90.5/5/2.5/2; v/v/v/v), yielding Ub$_{1-76}$ (K48 = L-azido-ornithine) 22. Subsequent Cu(I)AAC of alkyne 21 and azide 22 yielded non-hydrolyzable clicked Lys48 diUb 23 (Supplementary Fig. 10).

### Recombinant protein expression and purification of AMSH
**Protein expression constructs.** cDNA generated from MelJuSo cells was used as a PCR template for the cloning of AMSH. AMSH DNA was amplified by PCR reaction described in Supplementary Table 6 and 7 using the primers in Supplementary Table 8 and cloned into the pGEXNKI-GSThis3C-LIC vector using ligation-independent cloning[71]. In brief, a PCR fragment of AMSH flanked with specific LIC sequences was generated, and the pGEXNKI-GSThis3C-LIC vector was cleaved with KpnI enzyme, followed by agarose gel extraction. Extracted PCR fragment and the vector were treated with T4 DNA polymerase in the presence of dATP and dTTP, respectively. The insert ligation into the vector was performed by mixing them in 2:1 (insert: vector) dilution and incubating them at room temperature for 5 min. Ligated DNA was transformed into DH5α *E. coli* strain. The expression construct was purified and confirmed by DNA sequencing.

**Expression and purification protocol of AMSH.** The GST-tagged AMSH construct was transformed into the Rosetta *E. coli* strain. A single colony was grown in LB media containing 100 µg/µL ampicillin and 34 µg/µL chloramphenicol at 37 °C overnight as a starter culture. The starter culture was diluted in LB media containing 100 µg/µL ampicillin and 34 µg/µL chloramphenicol for a large-scale protein expression. The bacterial culture was incubated at 37 °C until A600 reached 0.6-0.8. Further, the culture was incubated at 30 °C for

4 hours after adding a 500 μM final concentration of IPTG (Isopropyl β-ᴅ-1-thiogalactopyranoside). Cells were lysed with lysis buffer (20 mM Tris, pH 8.0, 500 mM NaCl, 5 mM β-mercaptoethanol, and EDTA-Free complete protease inhibitor cocktail (Roche)) and sonication. The lysates were centrifuged at $21,000 \times g$ for 30 min at 4 °C. The supernatants were incubated with pre-washed Glutathione Sepharose™ 4 Fast Flow (GE Healthcare, Cat. 17-5132-03) for 1 hour at 4 °C under gentle rotation, and the beads were then washed with wash buffer (20 mM Tris, pH 8.0, 500 mM NaCl, 5 mM β-mercaptoethanol). Protein was eluted with elution buffer containing 20 mM Tris, pH 8.0, 500 mM NaCl, 5 mM β-mercaptoethanol, and 20 mM Glutathione. After elution, the GST tag was removed using 3C protease under dialysis against the buffer containing 20 mM Tris, pH 8.0, 500 mM NaCl, 5 mM β-mercaptoethanol and the protein was further purified on a size exclusion column (S200 16/60 column) using an ÅktaPrime (GE Healthcare) purifier. All proteins were aliquoted and stored at −80 °C.

### General method SDS-PAGE analysis

After indicated reaction time, the reaction was quenched by addition of 3x reducing sample buffer (SB) (containing 900 μL 4x LDS sample buffer (NuPAGE, Invitrogen) diluted with 210 μL water and 90 μL β-mercaptoethanol) and heated to 95 °C for 5 min. (denaturing conditions). Samples were loaded on precast 12% NuPAGE® Novex® Bis-Tris Mini Gels (Invitrogen) and resolved by SDS-PAGE gel electrophoresis using MES running buffer (NuPAGE MES SDS running buffer 20X, Novex by Life Technologies). Reference protein standard/ladder: SeeBlue™ Plus2 Pre-stained Protein Standard (Invitrogen, cat# LC5925). Proteins were visualized by InstantBlue™ (Expedeon Protein Solutions, #ISB1L), and stained gels were scanned using a GE Healthcare Amersham Imager 600.

### Characterization of all eight neutron-encoded diUb isoforms, K48 click diUb and Ub$_{1-74}$ by SDS-PAGE analysis

All eight neutron-encoded diUb isoforms, K48 click diUb and Ub$_{1-74}$ stock solutions were diluted to ~3.5 μM in a buffer containing 50 mM Tris•HCl, 100 mM NaCl, pH 7.55. To 40 μL of these solutions 20 μL 3x SB was added, samples were boiled and loaded on gel (15 μL/lane) and separated by gel electrophoresis. (Fig. 2d and Supplementary Fig. 2).

### SDS-PAGE analysis of USP21 mediated hydrolysis of neutron-encoded diUb

All neutron-encoded diUb, K48 click diUb and Ub$_{1-74}$ stocks were separately diluted to 2x final concentration (5 μM) in a buffer containing 50 mM Tris•HCl, 20 mM NaCl, pH 7.55, 5 mM DTT. USP21[196-565] was diluted to 2x final concentration (150 nM) in a buffer containing 50 mM Tris•HCl, 20 mM NaCl, pH 7.55, 5 mM DTT. Subsequently, 15 μL of diUb solution (5.0 μM) was mixed with 15 μL of enzyme solution (150 nM). Hydrolysis reactions were incubated for 3 hours at 37 °C. Samples for timepoint 0 min. were quenched before hydrolysis reaction started by mixing a diUb solution (2x final concentration, 5 μL) with 3x SB (5 μL) and subsequent addition of DUB (2x final concentration, 5μL).

Next to that, all neutron-encoded diUb stocks were mixed to yield a solution containing 2.5 μM of all eight neutron-encoded diUbs (1.5x final concentration) in a buffer containing 50 mM Tris•HCl, 20 mM NaCl, pH 7.55. USP21[196-565] was diluted to 3x final concentration (1200 nM) in a buffer containing 50 mM Tris•HCl, 20 mM NaCl, pH 7.55, 5 mM DTT. Subsequently, 15 μL of the 8x neutron-encoded diUb solution (8x 2.5 μM) was mixed with 7.5 μL of enzyme solution (1200 nM). Samples for timepoint 0 minutes were quenched before hydrolysis reaction started by mixing a diUb solution (1.5x final concentration, 5 μL) with 3x SB (5 μL) and subsequent addition of DUB (2x final concentration, 2.5 μL).

Samples from the reaction mixture (10 μL) were taken after 180 min. and the reaction was quenched 3x SB (5 μL) and analyzed

according to the general method for SDS-PAGE analysis (Fig. 3b and Supplementary Fig. 4).

### SDS-PAGE analysis of OTUB1, OTUD1 and USP21 mediated hydrolysis of synthetic neutron-encoded and enzymatically prepared Lys48-, Lys63- and Met1-linked diUb

Enzymatically prepared Lys48- and Lys63-linked diUb were obtained from Ubiquigent #60-0106-050 and #60-0107-010. Enzymatically prepared Met1-linked diUb was prepared in house[40].

Synthetic neutron-encoded and enzymatically prepared Lys48-, Lys63- and Met1-linked diUb stocks were diluted to 2x final concentration (approx. 10 μM, diUb conc. equalized by SDS-PAGE and InstantBlue™ staining) in a buffer containing 50 mM Tris•HCl, 100 mM NaCl, pH 7.55, 10 mM DTT. OTUB1 (FL), OTUD1[287-481] and USP21[196-565] were diluted to 2x final concentration (3.2 μM, 0.2 μM and 150 nM respectively) in a buffer containing 50 mM Tris•HCl, 100 mM NaCl, pH 7.55, 10 mM DTT. Subsequently, 20 μL of diUb solution (-10 μM) was mixed with 20 μL of enzyme solution (3.2 μM, 0.2 μM or 150 nM). Reactions were incubated for 30 minutes at 37 °C. Samples for timepoint 0 min. were quenched before hydrolysis reaction started by mixing a diUb solution (2x final concentration, 5 μL) with 3x SB (5 μL) and subsequent addition of DUB (2x final concentration, 5μL). Samples from the reaction mixture (10 μL) were taken after 10 and 30 min. and the reaction was quenched with 3x SB (5 μL) and analyzed according to the general method for SDS-PAGE analysis. (Supplementary Figs. 5 and 6).

### In vitro DUB assays with mass spectrometry read-out

The assays were performed in a 1.5 mL Eppendorf tube at 37 °C in a buffer containing 50 mM Tris•HCl, 20 mM NaCl, pH 7.6 and 5.0 mM final concentration of DTT or TCEP.

All eight neutron-encoded diubiquitins were mixed in an equimolar amount (8x 2.5 μM; 1.5x final concentration). Neutron-encoded K6 (145.5 μL), K11 (142.5 μL), K27 (68.25 μL), K29 (72.25 μL), K33 (178.5 μL), K48 (247.5 μL), K63 (321.75 μL) and M1 (139.5 μL) stocks were mixed and diluted with buffer (50 mM Tris•HCl, 20 mM NaCl, pH 7.6; 183.75 μL). Recombinant purified DUBs were obtained from commercial sources, received as a gift, or expressed and purified according to reported procedures (details provided in Supplementary Table 2). Recombinant purified DUBs were diluted to 3x final concentration (12 or 6 μM, 1,2 μM, 0,12 μM and 0,012 μM respectively) in a buffer containing 50 mM Tris•HCl, 20 mM NaCl, pH 7.6 and 15 mM DTT or TCEP. The diUb mixture (15 μL) was added to the Eppendorf tube and 1.33 μL was taken for timepoint 0. Subsequently, the enzyme (6.84 μL, 3x final concentration) was added to the remaining diUb mixture (13.67 μL, 1.5x final concentration). The reaction mixtures were incubated at 37 °C for 180 min. After 2, 5, 10, 30, 60 and 180 min a sample was taken from the reaction mixture for analysis.

Sample preparation: After indicated incubation time, 2 μL of the reaction mixture was taken. The reaction was quenched by acidification of the mixture and internal standards for MS analysis were directly added. Therefore, 2 μL of the reaction mixture was quenched, spiked and diluted with a mixture containing 0.4μL 10% TFA in MQ, 1.6 μL 0.1% FA in MQ and 12 μL of the internal standard solution (1 μL of 5.0 μM internal standard diluted with 11 μL of 0.1% FA in MQ).

Samples were collected in a 96-well plate and measured by LC-MS analysis. 8 μL of the sample was injected onto the column.

Data acquisition: The samples were separated by an Acquity H-class UPLC system using a BEH C4 column (300Å, 1.7 μM (2.1 x 50 mm)), column $T = 60$ °C. For the first 2.5 min, the flow was diverted from the detector to flush the column with 2% ACN in H$_2$O and 0.1% FA at 1 mL/min to elute most of the buffer components and salt. After 2.5 min. proteins were eluted using a shallow gradient that ranged from 26% up to 30% ACN in H$_2$O with 0.1% FA over 1 min using a flow rate of 0.6 mL/min, which was able to separate monoUb and diUb products

(baseline level). Products were analysed by intact MS analysis on a XEVO G2 XS Q-TOF in reflector positive ion mode with a resolution of $R = 22,000$ using positive electrospray ionization (ESI) (Cap. V. = 0.5 kV) and a detection range of $m/z$ 550-2000. The system check of the detector voltage, lock mass accuracy (of LeuEnk) and calibration using NaI solution were performed daily prior to analysis. Lock mass correction was applied during each analysis, to correct for possible mass shifts during the course of the assay.

Data analysis: Proteowizard 3.0.20274 was used to convert raw data files to the mzXML file format[72]. mzXML files were further processed using LaCyTools version 22.04.29. The alignment was performed using theoretical $m/z$ values of various charge states from the internal standard and the expected elution time (Supplementary Table 3). The extraction parameters are specified in Supplementary Table 4. Atom compositions were determined of all eight diUb molecules, all nine monoUbs that can be formed, $Ub_{1-74}$ as well as the non-hydrolysable clicked K48 diUb and listed as analytes (Supplementary Table 5). From all analytes, their retention time was predicted and the charge states that should be taken along during quantification are specified (Supplementary Table 5). The LaCyTools output file (Summary.txt) was further processed in Microsoft Excel, where boundaries for quality control parameters (Mass Accuracy < 15 ppm; IPQ < 0.25; S/N > 9) were set and the areas of all $m/z$ peaks, within the quality control boundaries, from the same analyte were summed.

The absolute area for each analyte was normalized at each time-point using the internal standard, non-hydrolysable clicked Lys48 diUb. The normalized area under the curve for each analyte was set to 100% remaining diUb at $t = 0$ and percentage remaining diUb at each timepoint was calculated relative to this, to account for variations in ionization of different diUb isoforms. Percentage remaining diUb was plotted against time using GraphPadPrism 9.3.0.

For the monoUb signals, the total absolute area under the curve was normalized at each timepoint using the internal standard $Ub_{1-74}$. The concentration of monoUb present at each timepoint was calculated using the theoretical concentration of present $Ub_{1-74}$ in the analysed mixture. Concentration monoUb was plotted against time using GraphPadPrism 9.3.0.

### Assay window and linearity determination

The assay concentration window is considered to be valid at concentrations for which the amount of loaded diUb corresponds to the experimentally determined amount. Linearity curves (theoretical amount of diUb plotted against determined amount) were constructed for all eight diUbs over the concentration range of 0.0-2.0 μM in a mixture containing all diUbs and K48 click as internal standard in three separately performed experiments on different days.

All eight diUb solutions were mixed (final concentration 2.0 μM of all linkages) and a serial dilution of 0.00, 0.25, 0.50, 0.75, 1.00, 1.25, 1.50, 1.75 and 2.00 μM was made in buffer (50 mM Tris, 20 mM NaCl). Each sample (2 μL) was diluted with 14 μL of a mixture containing 0.4μL 10% TFA in MQ, 1.6 μL 0.1% FA in MQ and 12 μL of the internal standard solution (1 μL of 5.0 μM internal standard diluted with 11 μL of 0.1% FA in MQ). A total of 8 μL of these sample mixtures were loaded and analyzed in an identical way as during the DUB assays. The data was quantified with LaCyTools. Non-hydrolysable clicked K48 click area was linked to the K48 click diUb concentration present in the mixture, using the measured area of all eight diUbs, their measured concentration was calculated. The theoretical amount of all eight diUbs present was plotted versus the measured and calculated amount of all eight diUbs. The concentration window was determined to be valid between 0.5 and 2.0 μM (Supplementary Fig. 7).

For the assay, a starting concentration of 1.6 μM of each diUb linkage was chosen. Assuming the linear detection threshold of monoUb and diUb are similar, it was presumed that as soon as the concentration of present diUb molecules dropped below the linear detection range, the concentration of present monoUb isoforms reached the linear detection threshold. Resulting in a reliable read-out of at least one of the analytes types during the assay. A higher starting concentration was avoided to prevent potential detector crowding or overloading of the LC column.

### Reporting summary

Further information on research design is available in the Nature Portfolio Reporting Summary linked to this article.

## Data availability

The experimental data that support the findings of this study are available from the corresponding author upon request. All MS datasets generated and/or analysed during the current study have been deposited with the ProteomeXchange Consortium via the MassIVE partner repository with data set identifier MSV000090455 [https://doi.org/10.25345/C5BN9X72V]. Full gel images and biochemical assay readings are provided in the Source Data file. The synthetic neutron-encoded diUb reagents and internal standards will be made available by the authors upon request. Source data are provided with this paper.

## Code availability

LaCyTools is freely available for download at https://github.com/Tarskin/LaCyTools.

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

## Acknowledgements

This work is dedicated to the memory of Prof. Dr. Huib Ovaa. We thank Alfred Vertegaal for the scientific discussions and Dris el Atmioui for his assistance with solid-phase peptide synthesis. Patrick Celie, at The Netherlands Cancer Institute (NKI) Protein Facility, is thanked for the expression and purification of recombinant DUBs. Expression constructs for USP7 and USP11 were kind gifts from Titia K. Sixma, and the expression construct for USP9X was a kind gift from Yufeng Tong. This work was supported by a VICI grant (no. 724.013.002) to H.O., a VENI and VIDI grant (no. 722.014.002 and no. VI.Vidi.192.011) to G.J.v.d.H.v.N. from The Netherlands Organization for Scientific Research (NWO) and an ICI grant (no. ICI00026) to A.S.

## Author contributions

The concept of this study was conceived by P.P.G. and H.O. Ub analogues were designed by P.P.G. Synthesis of Ub analogues was executed by B.D.M.v.T. with technical assistance from C.M.P.T.O. and P.J.M.H. under supervision of G.J.v.d.H.v.N. and P.P.G. Biochemical assays were performed by B.D.M.v.T. Data analysis was performed by B.D.M.v.T. with the assistance of G.S.M.L.-K. Data analysis software was adjusted by B.C.J. DUBs were expressed by R.Q.K., A.M. and A.S. Plasmid design and DUB expression by D.K. Mass spectrometry support and gradient design were done by B.R.v.D. The manuscript was prepared by B.D.M.v.T., G.J.v.d.H.v.N. and P.P.G. with input from all authors. The project was supervised by G.J.v.d.H.v.N., M.W., H.O. and P.P.G.

## Competing interests

The authors declare no competing interests.
