## [Peer Review File · Nature Communications]

REVIEWER COMMENTS

Reviewer #1 (Remarks to the Author):

Deubiquitinating enzymes (DUBs) are key regulators in the ubiquitin (Ub) system via reversely disassembly poly-ubiquitin chains. Current technologies towards deciphering DUB's selectivity over eight di-Ubs mainly rely on SDS-PAGE, in which di-Ubs cleavage were conducted individually. However, in cells, all linkages coexist and may influence each other. Therefore, it is of value to study DUB's linkage selectivity under a more biologically relevant context where all eight types di-Ub are present in a single mixture.

In this manuscript, Geurink et al. reported a novel multiplexed mass spectrometry-based DUB assay to analyze DUB specificity where all eight di-Ubs coexist. To achieve this, the authors chemically synthesized all eight different ¹³C and ¹⁵N labelled di-Ubs, each bearing a distinguishable molecular weight, allowing for discrimination between each of the eight di-Ubs and nine monoUb in mass spectra. The authors demonstrated the utility of the strategy by profiling the linkage specificity of 20 DUBs, and provided some new insights into DUB's activity, for example, the authors found that some DUBs of USP family harboring a concentration dependent linkage specificity for the first time.

In conclusion, this work highlights chemical protein synthesis as an effective route to obtain well-designed Ub tools to study the Ub code. The reported strategy would undoubtedly benefit the field of DUB's activity and selectivity research. Therefore, this work deserve a publication in Nature Communications if the authors can address the following questions.

(1) In this work, the authors chose ¹³C and ¹⁵N labelled amino acids Val, Leu, and Ile to introduce mass differences among di-Ubs. Is there any criteria to select which I/L/V to isotopically label for the proximal Ub of a given di-Ub, as there are 7 Ile, 9 Leu and 4 Val in Ub. For example, why the author chose to label V70 and L73 for K6-diUb.

(2) Have the authors tested the selectivity and efficiency of DUB for chemically synthesized di-Ub versus enzymatically prepared di-Ub?

(3) In this work, the authors chose five UbiCRest DUBs (which are valuable tools to analyze Ub chain linkage) to examine their selectivity. However, the previously reported K48-linkage specific DUB

OTUB1 and K11-specific DUB Cezanne were omitted. These two DUB are also recommended to be introduced to probe if there are some discrepancies when eight di-Ub coexist.

(4) The authors found that AMSH-LP had a broad specificity, which is inconsistent with previous results, and in the Discussion, the author described that “While comparing our data to literature findings, the most striking results were obtained for metallo-DUB AMSH-LP as, despite being annotated as a highly Lys63 specific DUB, our data revealed full linkage promiscuity in the presence of all diUb linkages and this effect is not dose-dependent”, this is not exactly correct as in Ref 26 (Nature communications 2014) and 62 (Nature 2008) they used truncated constructs of AMSH-LP, whereas in this work, the author used full length enzyme.

(5) In the Discussion, it is written “We also showed a first indication that different diUb linkage types can influence each other's cleavage rates and that different diUb linkages are sometimes processed in a specific order.” Have the authors found that one di-Ub can activate or inhibit the hydrolysis of another di-Ub by DUB? Is there any data support that “different diUb linkage types can influence each other's cleavage rates”? This manuscript seems do not have any kinetic data on di-Ubs cleaved individually.

(6) In the Discussion, it is written “Another interesting finding is the high preference of the metallo-DUB RPN11/RPN8 complex for Lys11 diUb, instead of its reported linkage promiscuity. This observed Lys11 preference may be explained by a higher catalytic efficiency for Lys11 chains, or because of the presence of other diUb linkages.” In fact, in Ref 54(NSMB 2014), Martin et al. have shown that Rpn8/11 have a preference for K11-diUb, and they find this comes from a higher Kcat for K11-diUb.

(7) Can this method be generalized to longer and more complex Ub chains? In cells, there exist tri-, tetra-, penta-, (even longer) and branched ubiquitin chains and can be reversibly regulated by DUBs.

Reviewer #2 (Remarks to the Author):

By removing ubiquitin modifications and dismantling ubiquitin chains, deubiquitinases (DUBs) regulate nearly every facet of eukaryotic biology. Defining the substrate specificities of DUBs is thus necessary to better understand their function. In this manuscript, Geurink and co-workers synthesize a series of di-ubiquitin isotopologues to probe the linkage specificities of DUBs using intact MS. In principle, there are two major advantages relative to existing methods: 1) each di-ubiquitin isomer is

in its native form, i.e., no additional tags have been added to monitor the cleavage process; and 2) competition between different di-ubiquitin isomers can be accounted for. There are, however, some major concerns:

- Ubiquitin chains vary in length and morphology and both factors affect DUB activity; however, this method does not account for such complexity. Nor does it account for the effects of a ubiquitinated substrate on activity. Thus, the results obtained from this method might not be relevant to the actual biology of a particular DUB.
- The method is not quantitative, which precludes steady-state kinetic analyses of reactions with DUBs that are either indifferent or sensitive to the presence of other isomers.
- Figures 4c, 5c, and Supplementary Figure 6 show conversion vs. time profiles in which the % remaining reaches zero in many cases. There are also several instances in the main text in which the authors state “full conversion” has been achieved. How can this be when the dynamic range is quite limited (1.6 to 0.8 μM)?
- Based on the standard curves shown in Supplementary Figure 4 the signal does not appear to be very responsive to changes in concentration for most of the di-ubiquitin isomers except for K6 and K11. Even for the K6 and K11 dimers, the data looks nonlinear. These data raise concerns about how the quantitative analysis was performed.

Rebuttal to reviewers' comments manuscript NCOMMS-22-22834: "Competition or indifference: Neutron-Encoded Ubiquitins to profile linkage selectivity of deubiquitinating enzymes"

Before providing a point-by-point response to the reviewers' comments, we first want to thank both reviewers for their critical and comprehensive assessment of our manuscript. The points raised by the reviewers were highly useful to improve our manuscript and in some cases revealed that the way we presented our data was not always clear to the reader. We like to point out that a few points raised by reviewer 2 most likely originate from a misinterpretation of the linearity data presented in the Supplementary Information. We now realize that the way we initially presented the linearity data could have been confusing. We opted to show the linearity for 2 of the diUb reagents (K6 and K11), while showing in the same figure the lines for all other diUb reagents, although these were not present in the analysed sample. The reason for this was to show that the absent diUb reagents were indeed not detected. After reading the reviewer comments we decided to redo the linearity determination but this time in the presence of all diUb reagents. This is included as the new SI Figure 6, which now convincingly shows a proper linear behaviour of all reagents above 0.5 μM .

We performed the following additional experiments:

1. We determined the diUb selectivity for OTUB1 at four different enzyme concentrations. The outcome was that OTUB1 showed a clear Lys48 specificity, as expected. (Figure 5 & SI Figure 8a)
2. We determined the diUb selectivity for Cezanne at four different enzyme concentrations. The outcome was that Cezanne is Lys11 specific at lower concentration (0.04 μM) but loses this specificity at higher enzyme concentrations, as it processed Lys6, Lys48 and Lys63 at 0.4 μM and 4.0 μM as well. This is in line with reported data. (Figure 5 & SI Figure 8a)
3. We compared the processing of synthetic and enzymatically prepared Lys48-, Lys63- and Met1-linked diUb by OTUB1, OTUD1 and USP21 using SDS-PAGE analysis. The outcome was that there were no difference in the DUB selectivity of diUb cleavage between synthetic and enzymatically prepared diUbs, and the enzymatically prepared diUbs were processed slightly faster than the synthetic diUbs. (SI Figures 4 & 5)
4. We measured the linearity curves for all eight diUbs (0 – 2.0 μM) present in the mixture. The outcome was that each linkage showed a linear behaviour for the theoretical versus the measured concentration diUb above 0.5 μM . (SI Figure 6)

In addition, we wrote an additional paragraph in the Discussion section on the biological relevance of our approach, including a discussion about the current limitations and the potential for further development. All changes made to the main text of the manuscript are highlighted in yellow.

REVIEWER COMMENTS

Reviewer #1:

(1) In this work, the authors chose ^{13}C and ^{15}N labelled amino acids Val, Leu, and Ile to introduce mass differences among di-Ubs. Is there any criteria to select which I/L/V to isotopically label for the proximal Ub of a given di-Ub, as there are 7 Ile, 9 Leu and 4 Val in Ub. For example, why the author chose to label V70 and L73 for K6-diUb.

RESPONSE: In principle, for the designed assay it does not matter at which locations the isotope labelled amino acids are introduced, since we measure intact proteins. It is only important that the correct mass difference is introduced and that the mass difference between the different diUb chains is sufficient to distinguish them from each other by mass spectrometry. Also, for the selectivity of DUBs it does not matter at which location the mass difference is introduced, since the native sequence is maintained. All neutron-encoded amino acids were incorporated in the proximal Ub only, so the isotope code could still be tracked by MS upon cleavage of the diUb substrates into the monoUb products. This gave us a high degree of flexibility to choose the optimal positions within the sequence of the proximal Ub. Our choice was made based on the most optimal way for solid phase peptide synthesis (SPPS). Some Leucine and Isoleucine amino acids are introduced as part of a pseudoproline building block (e.g. Leu8, Ile 13, and Leu56). Since neutron-encoded pseudoproline building blocks are not commercially available these three positions could not be used. We chose to use predominantly the amino acid positions closer to the resin, at the early stage of the SPPS when the growing peptide chain is still relatively short, because the amino acid coupling reactions are generally more efficient there. This way we were able to use smaller amounts of the isotope-labelled amino acids and still achieve full conversion, which lowered the required amounts of these relatively expensive building blocks. Finally, we tried to equalize the required amounts of the three isotope-labelled amino acids as much as possible, which for example led to the choice of Val70 instead of Leu71 in the case of Lys6 diUb.

In order to clarify our choices a little better we added the following sentence to the main text (Page4, lines 102-104): [This way each linkage could be identified by MS in both the diUb (substrate) as well as the monoUb (product) form (*vide infra*). Besides, since the assay relies on intact mass analysis only, the read-out is unaffected by the exact location of heavy-isotope introduction.

(2) Have the authors tested the selectivity and efficiency of DUB for chemically synthesized di-Ub versus enzymatically prepared di-Ub?

RESPONSE: We did not test this before but such data will further validate the use of our synthetic constructs. We therefore purchased enzymatically prepared Lys48- and Lys63-linked diUb from Ubiquigent and incubated these alongside the in-house available expressed linear (M1) diUb with three DUBs (USP21, OTUB1 and OTUD1) at previously reported concentrations (Faesen *et al.* Chem. Biol. 2011 and Mevissen *et al.* Cell 2013). This was done side-by-side with the corresponding synthetic neutron-encoded diUbs. The cleavage was followed by SDS-PAGE analysis after 0, 10 and 30 minutes. The outcome was that the linkage specificities were retained (e.g. USP21 cleaves all three, OTUB1 only cleaves Lys48 and OTUD1 only cleaves Lys63) and the enzymatically prepared diUb were processed slightly faster by OTUB1 and OTUD1, compared to our synthetic material,

although the differences were small. The gel images were included in the supporting information (**Supplementary Figure 4 and 5**), we updated the methods section (Page 21, lines 600-615) and we added the following sentences to the main text (Page 5, lines 125-130): “The diUb integrity of our synthetic constructs was confirmed in an assay where we compared the cleavage efficiency of neutron-encoded Lys48-, Lys63- and Met1-linked diUbs with their corresponding enzymatically prepared diUbs side-by-side using OTUB1 (reported Lys48 specific)²⁴, OTUD1 (reported Lys63 specific)²⁴ and USP21 (reported unspecific)²⁵. DUB mediated hydrolysis was analysed by SDS-PAGE (**Supplementary Figure 4 and 5**), which showed that the enzymatically prepared material and the synthetic material were processed comparably.”.

(3) In this work, the authors chose five UbiCRest DUBs (which are valuable tools to analyze Ub chain linkage) to examine their selectivity. However, the previously reported K48-linkage specific DUB OTUB1 and K11-specific DUB Cezanne were omitted. These two DUB are also recommended to be introduced to probe if there are some discrepancies when eight di-Ub coexist.

RESPONSE: We performed the suggested assays and included the data in the main manuscript. The outcome was that OTUB1 mainly cleaves Lys48 and we observe minor processing of Lys63 and M1 diUb at higher concentrations, which corroborates literature findings. Cezanne shows Lys11 specificity as expected and becomes linkage-promiscuous at higher concentrations. Based on this, we updated the following sections: We replaced 20 DUBs with 22 DUBs at various locations. Results (Page 6, lines 184-187; Page 6, lines 189-193; Page 7, lines 200-202), Discussion (Page 11, lines 322-326), **Figure 4b**, **Figure 5a** and **Supplementary Figure 8a**.

(4) The authors found that AMSH-LP had a broad specificity, which is inconsistent with previous results, and in the Discussion, the author described that “While comparing our data to literature findings, the most striking results were obtained for metallo-DUB AMSH-LP as, despite being annotated as a highly Lys63 specific DUB, our data revealed full linkage promiscuity in the presence of all diUb linkages and this effect is not dose-dependent”, this is not exactly correct as in Ref 26 (Nature communications 2014) and 62 (Nature 2008) they used truncated constructs of AMSH-LP, whereas in this work, the author used full length enzyme.

RESPONSE: The reviewer is correct that we used the full length enzyme and the reviewer also correctly states that in Ref 26 (Nature Communications 2014) the authors used a truncated version GST-tagged AMSH-LP(265–436) expressed in E. coli. We thank the reviewer for pointing this out since we accidentally overlooked this detail. However, in Ref. 62 (Nature 2008) in fact both the truncated version and the full length version of AMSH-LP were used although this does not become directly clear. In the main text of this paper only the truncated version is mentioned: “The region containing residues 264–436 of human AMSH-LP is sufficient for the Lys 63-linkage-specific DUB activity (Supplementary Fig. 1).” Supplementary Fig. 1a: “The AMSH-LP DUB domain can cleave Lys63-linked polyubiquitin chains but not Lys48-linked chains. Reaction mixtures were analyzed by SDS-PAGE after 20-hour reactions.” However, the full length version of AMSH-LP is shown in Supporting Figure 1b: “Full-length AMSH-LP can cleave Lys63-linked polyubiquitin chains but not Lys48-linked chains. In contrast, the USP family DUB, UBPY, can cleave both Lys48- and Lys63-linked polyubiquitin chains. Reaction mixtures were analyzed by SDS-PAGE after 20-hour reactions. “ So,

the authors of the Nature 2008 paper do test the full-length AMSH-LP but not on the entire linkage panel. In order to clarify and correct our statement we rewrote the corresponding part of the discussion (Page 12, lines 365-367), which now reads: “The DUB domain of AMSH-LP is being annotated as a highly Lys63 specific DUB^{26,66} and the full length enzyme is reported to cleave Lys63-linked polyUb chains, but not Lys48-linked polyUb chains.⁶⁶ Our data revealed full linkage promiscuity in the presence of all diUb linkages and this effect is not dose-dependent (**Fig. 6** and **Supplementary figure 8c**). “

(5) In the Discussion, it is written “We also showed a first indication that different diUb linkage types can influence each other's cleavage rates and that different diUb linkages are sometimes processed in a specific order.” Have the authors found that one di-Ub can activate or inhibit the hydrolysis of another di-Ub by DUB? Is there any data support that “different diUb linkage types can influence each other's cleavage rates”? This manuscript seems do not have any kinetic data on di-Ubs cleaved individually.”

RESPONSE: It is indeed the case that we do not have any kinetic data on individually cleaved diUbs and we do not have data that directly proves either the activation or the inhibition of the hydrolysis of one diUb linkage by another. It was also not the message that we intended with our statement. Our statement (that we presented as a ‘first indication’) was supposed to highlight the finding that diUb linkages are processed in a specific order, which is supported by the data for USP9x and USP10 (Page 7, lines 213-221). This could be the result of one or more diUbs influencing each other's cleavage rates (either inhibition or activation), but actual direct data to back-up the first part of the statement is indeed missing.

On the other hand, when comparing our data to published (kinetic) data of individual diUb linkages hydrolysis rates, there are in fact significant differences, which indirectly indicates that the presence of other or multiple linkages can lead to a change in hydrolysis rates of certain linkages.

For example, USP9x shows a faster processing of Lys63 compared with Lys11-linked diUb, according to Paudel *et al.* PNAS 2019. However, in our assay we observe faster processing of Lys11-linked diUb in comparison with Lys63-linked diUb, which could indicate that its catalytic efficiency in the presence of other linkages changes.

Furthermore, Faesen *et al.* Chem Biol 2011 showed that the catalytic activity of USP7FL is 1.5 times higher for Lys33- compared to Lys63- and Lys11-linked diUb. However, Lys11- and Lys63-linked diUb are processed faster in our assay (most clearly visible at 0.04 μ M USP7 FL) than Lys33-linked diUb.

For Cezanne (WT), the catalytic efficiency of Lys11-, Lys48- and Lys63-linked diUb is reported by Mevissen *et al.* Nature 2016 (Extended data 1c). A fourfold higher efficiency is observed for Lys63-over Lys48-linked diUb. Contrarily, in our assay, similar or better processing of Lys48-linked diUb is observed.

We added some lines in the Discussion section to better describe these findings. We also decided to remove the part of the sentence that contained the statement on the ‘influence of each other's cleavage rates’ in the summary of the Discussion section to better emphasize the actual finding on the specific cleavage order. This sentence has now been changed into (Page 13, lines 403-405): “Our data revealed that direct competition between different diUb linkages can alter a DUBs Ub chain

specificity pattern and we showed a first indication that different diUb linkages are sometimes processed in a specific order.”

The following sentences have been added to the Discussion (Page 11,12, lines 345-357): “Notably, since the reaction progress is being monitored at several timepoints (**Fig. 4a**), it is possible to deduce information on the cleavage rates of each diUb linkage type relative to each other. This provided some interesting observations that differ from literature findings. USP9x was earlier shown to faster process Lys63- compared to Lys11-linked diUb⁶⁴, whereas our data showed the exact opposite effect (**Supplementary Figure 8b**). Kinetic analysis of USP7 previously revealed a 1.5 times higher catalytic efficiency towards Lys33- over Lys11- and Lys63-linked diUb²⁵, which contrasts our finding that Lys33 is processed more slowly compared to Lys63 and Lys11 (**Supplementary figure 8b**). Also, our data revealed an similar processing rate of Lys48- and Lys63-linked diUb by Cezanne (**Supplementary figure 8a**), where a fourfold higher catalytic efficiency towards Lys63- over Lys48-linked diUb was reported previously.⁶⁵ These observed differences in hydrolysis rates between those reported for individual chains and the ones we found in the mixture containing all linkages may indicate that the presence of certain diUb linkages can lead to a change in hydrolysis rates of other linkages.”

(6) In the Discussion, it is written “Another interesting finding is the high preference of the metallo-DUB RPN11/RPN8 complex for Lys11 diUb, instead of its reported linkage promiscuity. This observed Lys11 preference may be explained by a higher catalytic efficiency for Lys11 chains, or because of the presence of other diUb linkages.” In fact, in Ref 54(NSMB 2014), Martin et al. have shown that Rpn8/11 have a preference for K11-diUb, and they find this comes from a higher Kcat for K11-diUb.

RESPONSE: Initially, we based our statement on the data presented in their SDS-PAGE analysis as shown in Figure 3b of the NSMB 2014 paper, where all isopeptide linked diUbs are taken along. However, we now had a careful look at their kinetics data presented in Table 2 of the NSMB 2014 paper, where the authors determined the kinetic parameters of K11, K48 and K63 diUb cleavage. From their data we calculated the catalytic efficiencies (k_{cat}/K_M) for the three linkages and this indeed results into a higher catalytic efficiency for K11 compared to K48 (~2 fold) and K63 (~4 fold) diUb. As such, we changed our statement by rephrasing the corresponding part in the Results (Page 8, lines 238-239) and Discussion (Page 12, lines 359-364).

(7) Can this method be generalized to longer and more complex Ub chains? In cells, there exist tri-, tetra-, penta-, (even longer) and branched ubiquitin chains and can be reversibly regulated by DUBs.

RESPONSE: Given the existence of more complex chains beyond diUb, this would indeed be a valuable extension of our method. It is beyond the scope of the current manuscript where we describe the development of this new type of assay, but the nature of our assay, the widely available chemical preparations of Ub constructs, as well as our MS-based read-out, should in principle allow for its application in the analysis of more complex Ub chains. To further discuss this topic we wrote a new paragraph in the Discussion section (Page 12,13, lines 374-397).

Reviewer #2:

(1) Ubiquitin chains vary in length and morphology and both factors affect DUB activity; however, this method does not account for such complexity. Nor does it account for the effects of a ubiquitinated substrate on activity. Thus, the results obtained from this method might not be relevant to the actual biology of a particular DUB.

RESPONSE: The reviewer is correct in stating that a wide variety of chains is present in cells that can affect DUB activity. Indeed, the assay in its current form has some limitations, as is the case for all other existing assays that address Ub linkage cleavage by DUBs. Notably, by introducing the element of linkage competition and by keeping the diUb native, we opted to mimic the native situation as good as possible. In the Ub field *in vitro* assays are commonly used and well accepted to characterize DUBs. They do not account for the full complexity of all possible Ub chains present in cells, but they are certainly considered to be relevant. Our method has several advantages over existing methods (such as the linkage competition element, the easy read-out, small amount of required substrates) but at the same time provides several options for improvements, such as the extension to more complex Ub chains and hybrid chains, which will bring the method closer to the actual biology. This comment from reviewer 2 is related to the comment from reviewer 1 about the possibility to extend our method to more complex Ub chains. We added a paragraph to the discussion section to address this topic.

(2) The method is not quantitative, which precludes steady-state kinetic analyses of reactions with DUBs that are either indifferent or sensitive to the presence of other isomers.

RESPONSE: We have the feeling that the reviewer got the wrong impression from our standard curves in Supplementary Figure 4, which we now realize is likely due to the confusing nature of this particular figure. This probably led to the conclusion that our method is not quantitative whereas in fact our method is quantitative and therefore does allow for steady-state kinetic analysis. This same confusion is likely behind the comment made at point 4 from this reviewer. We therefore address this topic in the answer to point 4.

(3) Figures 4c, 5c, and Supplementary Figure 6 show conversion vs. time profiles in which the % remaining reaches zero in many cases. There are also several instances in the main text in which the authors state "full conversion" has been achieved. How can this be when the dynamic range is quite limited (1.6 to 0.8 μM)?

RESPONSE: We consider both the diUb and the monoUb signals, which we attempted to explain in the Results section (Page 6, lines 179-180 "For data interpretation, both diUb consumption and monoUb appearance were taken into account.") and the Discussion (Page 10-11, lines 305-320 "In some cases....in our assay"). The moment the diUb signal drops below the detection limit we switch to the monoUb signal which is then well above the detection limit. We also explained this in the main text. Page 5, Line 149-155 in the main text states: "A linear response in concentration (and signal height) was well detected by the mass spectrometer between 0.5 μM and 2.0 μM of the diUbs and signals from diUb concentrations below 0.5 μM showed overlap with the background signals (**Supplementary Figure 6**). Therefore, a starting concentration of 1.6 μM of each diUb linkage was chosen to ensure a reliable read-out throughout the entire assay time as the concentration of at

least one of the analyte types, either monoUb or diUb, will always be above the linear detection threshold.” And Page 6, lines 174-177 in the main text states: “From the recorded data, diUb disappearance, as well as monoUb appearance, can be quantified, and these values should correspond with each other (Fig. 4c). The percentage of consumed diUb substrate and the concentration of formed monoUb were calculated and plotted over time for all measured DUBs at different concentrations (Supplementary Figure 8a-c).”

Meaning that from the range of 1.6 to 0.5 μM of diUb present in the sample, the quantified diUb is most reliable, however as soon as the concentration of diUb comes below 0.8 μM (50% conversion), the monoUb will rise above 0.8 μM and is more reliable to look at. Therefore one has to take both the diUb and the monoUb curves into consideration when interpreting the data.

Important to note here is that we reran the linearity measurements because of the comments from reviewer 2 (point 2 and 4). This led to a small change in our definition of the linear range, which had now been changed into 0.5 – 2.0 μM . This however does not change the outcomes of the study.

Our strategy is further illustrated by the following two examples:

OTUD3 specifically cleaved Lys6 and Lys11 diUb at 0.4 μM , confirming literature observations^{24,48,49} but interestingly also showed nearly full conversion of Lys48 and Lys63 diUb at 4.0 μM . (Page 7, lines 197-199)

At $t=180$ min, 25% of Lys63 and 7% of Lys48-linked diUb is remaining, meaning 0.4 μM and 0.112 μM respectively. So both below the linear detection range. However, one would expect that at $t=180$ min 1.2 μM of Lys63-related monoUb and 1.488 μM of Lys-48 related monoUb is formed if the diUb data is correct. Looking at the monoUb graph, one could see that 1.12 μM Lys63-related and 1.17 μM Lys48-related monoUb is formed, both concentrations fall into the linear detection range of monoUb. So 70% of Lys63 has to be consumed and 73% of Lys48 has to be converted to produce the monoUb molecules.

Strikingly, USP9x only started the consumption of Lys29 and Lys33 chains after Lys6, Lys11, Lys48 and Lys63 were almost fully processed (Page 7, lines 215-216).

The Lys6, Lys11 and the Lys48 diUb signal reached the baseline within 5 minutes after the assay started. In the monoUb graph you see that after 5 minutes, 1.3 μM of Lys6-related monoUb is formed, 0.8 μM of Lys11-related monoUb and 1.1 μM of Lys48-related monoUb. After 60 minutes a concentration of 1.6 μM for Lys6, 1.0 μM for Lys11 and 1.4 μM for Lys48 is reached, meaning 100%, 63% and 88% conversion relatively. Of note, the starting concentration of Lys11 was probably not exactly 1.6 μM but slightly lower, meaning an even higher conversion rate. 100% and 88% conversion are almost fully processed.

(4) Based on the standard curves shown in Supplementary Figure 4 the signal does not appear to be very responsive to changes in concentration for most of the di-ubiquitin isomers except for K6 and K11. Even for the K6 and K11 dimers, the data looks nonlinear. These data raise concerns about how the quantitative analysis was performed.

RESPONSE: We had a proper look at the original Supplementary Figure 4 and we realized that this figure likely has led to a misunderstanding. We apologize for the confusion we created with this figure. We performed some new measurements and re-created the figure, which we believe is now much more clear.

The original figure title implicated that all diUbs were present during the linearity measurement. This was however not the case, only Lys6 and Lys11 were present and we opted to show that the other linkages were indeed not detected by showing their corresponding lines in the same figure (which were around zero as expected). This probably led to the misinterpretation that the signal for the other linkages was not very responsive, whereas in fact, they were not present. In order to take away all confusion we performed the same linearity measurement but in this case included all eight diUb in the mixture. The outcome is a clear graph showing a proper linear behavior between measured and theoretical diUb amounts for the concentration range of 0.5 – 2.0 μ M. Compared to the previous experiment we now took along some more concentration points between 1 and 2 μ M. We therefore also changed the statement in the text about the linear range; this was 0.8 – 2.5 μ M but is now 0.5 – 2.0 μ M. Below 0.5 μ M the signal becomes too low to be accurately detected by the mass spectrometer. We replaced the original Supplementary Figure 4 by the new figure, which is Supplementary Figure 6 in the new version. We updated the methods section accordingly and made a few changes in the main text (Page 5, line 145 and lines 150-152). We hope that this clarification convinces the reviewer that our method is in fact quantitative, which should address point 2 and 4.

REVIEWERS' COMMENTS

Reviewer #1 (Remarks to the Author):

The authors have satisfactorily addressed the majority of my initial comments, the comparison of chemically and enzymatically synthesized diUbs towards DUBs, and addition of the OTUB1 and Cezanne substantially improved this work. Only a few minor points remain to be addressed before publication

1. I suggest authors verify their finding that AMSH-LP demonstrated a broad specificity in the presence of all diUb linkages, for example, by examine activity of AMSH-LP towards fluorescent labelled K48-diUb in the presence of other seven diUbs with an SDS-PAGE read-out .
2. Error bar should be included in all figures (main text and SI) if each experiment was conducted in 3 independent replicates.
3. In the revised manuscript, line 468 and 469, "500 mM IPTG" should be revised to 500 μ M.
4. In the revised manuscript, line 564 and 565, "m/z 550-2000" should be revised to m/z 550-2000 Da.

Reviewer #2 (Remarks to the Author):

In the revised manuscript the authors have addressed my primary concern, which was related to the standard curves in what is now Supp. Figure 6. The authors have also clarified how they are measuring conversion. I therefore recommend publication.

1. I suggest authors verify their finding that AMSH-P demonstrated a broad specificity in the presence of all diUb linkages, for example, by examine activity of AMSH-LP towards fluorescent labelled K48-diUb in the presence of other seven diUbs with an SDS-PAGE read-out.

We are grateful to the reviewer for suggesting this, as it resulted in the identification of a faulty enzyme stock, which I will explain here. We agreed with the reviewer that the suggested experiment would indeed be a good control experiment to perform. As such, we incubated AMSH-LP (same enzyme stock as used before) with the mixture of all 8 diUb linkages, 7 of which without a fluorescent label and 1 diUb linkage tagged with a fluorescent dye. We did this for 4 different linkages (K6, K33, K48 and K63) by making use of our previously developed diUb FRET pairs (Geurink et al. *Chembiochem* 2016, 17, 816-820). SDS-PAGE followed by fluorescence scanning of the gel indeed revealed processing of each of these fluorescent diUb linkages as shown in Appendix 1 at the end of this letter. Initially, we were quite satisfied with the results. However, upon close inspection of the Coomassie-stained gel, we observed many high MW protein bands, instead of one clear band belonging to AMSH-LP (Appendix 1, bottom panel, indicated in red). Hence we realized that something must be wrong with the enzyme stock solution. Indeed, upon close inspection we found that the assumed AMSH-LP stock was in fact not AMSH-LP but rather another DUB or a mixture of DUBs. This means that the results and conclusions we initially obtained for AMSH-LP were incorrect. This immediately turned our attention to carefully check and confirm the integrity and purity of all other DUBs in used our study. We therefore analyzed all used DUB stocks on gel (Appendix 2). Fortunately, all DUBs besides AMSH-LP mainly showed a single strong band at the correct molecular weight. For some DUBs, mainly USPs, some minor side bands are visible in addition to the main band, but this is quite common for these DUBs as it is also observed for the commercial USPs (see for example certificate of analysis for USP2 and USP8 from Ubiquigent: <https://www.ubiquigent.com/services/u-product/gst-usp2-50> and <https://www.ubiquigent.com/services/u-product/usp8>).

We therefore concluded that all experiments we included in our manuscript are good, except for AMSH-LP. We decided to purchase AMSH-LP from Abcam (<https://www.abcam.com/products/proteins-peptides/recombinant-human-amsh-lp-protein-ab139776.html>) and repeat the assay. This resulted in the expected outcome that AMSH-LP is specific for K63 diUb. We replaced the old AMSH-LP data with the correct new data and changed the appropriate text, figures and supplementary data accordingly.

We apologize for not checking the initially used AMSH-LP stock, and are grateful for the critical suggestions made by the reviewer to validate our results that led to identification of the initial error. We further confirm that all data that is now in the manuscript is correct. The main outcomes and

conclusions of our study still stand and the only consequence was that we had to take out the observation on AMSH-LP.

2. Error bar should be included in all figures (main text and SI) if each experiment was conducted in 3 independent replicates.

The linearity experiment from which the results are shown in Supplementary Figure 7 (SI Figure 6 in the previous version) is the only experiment that is conducted in 3 independent replicates. Here error bars are given with the corresponding correct description in the figure legend. The experiments from Figure 4, Figure 5 and Figure 6 are only conducted once with one DUB at one enzyme concentration. The only exception to this is the experiment studying USP9x in-house – 4.0 μ M that was performed in duplicate to show the repeatability of the assay. Data is depicted in two different graphs ($n=1$ and $n=2$) on page 15 of Supplementary Data 2. The study in this paper focuses mainly on method development followed by in-vitro validation of the assay on recombinant proteins. All the substrates are synthesized in a single batch per substrate, and only one enzyme batch was available for this study (unless stated otherwise). Replicates with the same substrate and the same enzyme batch would merely be technical replicates and no true biological replicates. As stated in the author checklist statistics derived from technical replicates is discouraged. In addition: most studied enzymes are previously studied in literature and the measured data corresponds with findings from literature.

3. In the revised manuscript, line 468 and 469, “500 mM IPTG” should be revised to 500 μ M. We thank the reviewer for this comment.

This was a typo and we corrected this in the revised manuscript.

4. In the revised manuscript, line 564 and 565, “m/z 550-2000” should be revised to m/z 550-2000 Da.

We did not change this because the unit of m/z is not Da. m/z is the mass of an analyte (with unit Da) divided by the charge. It is not common to give a unit for an m/z value or range.

Appendix 1

SDS-PAGE analysis of diUb cleavage by AMSH-LP. Read-out by fluorescence scanning (top) and Coomassie staining (bottom). AMSH-LP was incubated with a mixture of all 8 diUb linkages consisting of 7 untagged diUbs and 1 fluorescently labelled diUb (FRET, Geurink et al. *Chembiochem* 2016, 17, 816-820) as indicated. The Coomassie staining showed many unexpected high MW bands that do not originate from AMSH-LP. The conclusion was that this DUB is not AMSH-LP but another DUB, that is not specific for any diUb linkage.

Fluorescence scan

Coomassie stain

Appendix 2

SDS-PAGE analysis with InstantBlue staining of DUBs used in the current study.